# Pull Requests as a Training Signal for Repo-Level Code Editing

Qinglin Zhu [* 1]   Tianyu Chen [2]   Shuai Lu [2]   Lei Ji [2]   Runcong Zhao [1]   Murong Ma [3]   Xiangxiang Dai [4]
Yulan He [1 5]   Lin Gui [1]   Peng Cheng [2]   Yeyun Gong [2]

## Abstract

Repository-level code editing requires models to understand complex dependencies and execute precise multi-file modifications across a large codebase. While recent gains on SWE-bench rely heavily on complex agent scaffolding, it remains unclear how much of this capability can be internalised via high-quality training signals. Thus, we propose **Clean Pull Request (Clean-PR)**, a mid-training paradigm that leverages real-world GitHub pull requests as a training signal for repository-level editing. We introduce a scalable pipeline that converts noisy PR diffs into Search/Replace edit blocks by reconstruction and validation, yielding the largest publicly available corpus of **2M PRs** across **12 languages**. Leveraging this signal, we implement a mid-training stage, followed by an Agentless-aligned SFT with error-driven augmentation. On SWE-bench, our model significantly outperforms the instruct model baseline, achieving absolute gains of 13.6% on Lite and 12.3% on Verified. Our results show that repo-level capabilities can be internalised into model weights under a simplified Agentless protocol, reducing reliance on heavy inference scaffolding.

## 1. Introduction

Repository-level software engineering (SWE) has emerged as a crucial testbed for code-capable large language models (LLMs), driven by executable benchmarks such as SWE-bench (Jimenez et al., 2024). State-of-the-art SWE systems typically rely on composite architectures, combining agentic tool use (Yang et al., 2024; Wang et al., 2025), structured localisation (Xia et al., 2025; Jiang et al., 2025), and extensive test-time scaling (Antoniades et al., 2025). While effective, this complexity makes it difficult to attribute performance gains to any single factor. This motivates a fundamental question: *how much repository-level editing capability can be encoded directly into model weights?*

Answering this question requires abundant, high-quality training data aligned with repository-level editing. However, as summarised in Table 1, current data landscapes present a dichotomy. On one hand, SWE-bench-style datasets provide high-fidelity, executable verification but are expensive to curate and limited in scale (Pan et al., 2025; Jain et al., 2025). On the other hand, massive pretraining corpora like The Stack (Kocetkov et al., 2023) and CodeReview (Li et al., 2022) offer scale but lack instruction on *how* to modify codebases to resolve issues. There remains a clear gap in datasets that combine the scale of natural corpora with the structured, multi-file editing signals required for SWE tasks

Open-source Pull Requests (PRs) offer a promising middle ground, naturally coupling human intent expressed in natural language (descriptions, discussions) with accepted code changes, thereby providing a rich training signal for both *where to edit* and *how to edit*. Yet, raw PR traces (visualised in Table 13) are a noisy proxy for high-quality training data. As shown in Table 2, a naïve ingestion of GitHub PRs results in substantial noise: a significant majority are discarded due to being bot-generated, lacking core source code changes, or simply remaining unmerged. This motivates a rigorous conversion pipeline that turns PRs into model-ready and scalable training data for repository-level editing.

In this work, we propose **Clean-PR**, a scalable mid-training paradigm that transforms noisy PRs into a rigorous training signal for repository-level editing (Figure 1). To ensure high-quality supervision at scale, we implement a data construction pipeline consisting of three main steps. **First, Noise Filtering and Issue Linking:** We design a high-precision pipeline to filter low-signal PRs and, given that PR and issue content are stored separately, we detect referenced identifiers to augment examples with faithful issue context. **Second, Search/Replace Conversion:** We reconstruct repository-consistent *before/after* states and *verify* that the derived Search/Replace blocks deterministically re-

---

*Work done during internship at Microsoft Research. Qinglin Zhu is supervised by Lin Gui and Yulan He.    [1]King's College London, UK [2]Microsoft Research Asia, CN [3]National University of Singapore, SG [4]Chinese University of Hong Kong, HK [5]The Alan Turing Institute, UK. Correspondence to: Tianyu Chen <chentianyu@microsoft.com>.

*Proceedings of the 43rd International Conference on Machine Learning*, Seoul, South Korea. PMLR 306, 2026. Copyright 2026 by the author(s).

*Table 1.* Comparison of repo-level SWE datasets and natural code-edit corpora.

| Dataset | Real task | Multi-file | Diff format | #Lang | #Repo | #Instances |
|---|---|---|---|---|---|---|
| *SWE-bench-style datasets:* | | | | | | |
| Multi-SWE-bench (Zan et al., 2025) | ✓ | ✓ | Diff | 7 | 39 | 1,632 |
| SWE-PolyBench (Rashid et al., 2025) | ✓ | ✓ | Diff | 4 | 21 | 2,110 |
| SWE-bench-Live (Zhang et al., 2025) | ✓ | ✓ | Diff | 1 | 164 | 1,565 |
| SWE-Gym (Pan et al., 2025) | ✓ | ✓ | Diff | 1 | 11 | 2,438 |
| R2E-Gym (Jain et al., 2025) | ✓ | ✓ | Diff | 1 | 10 | 8,135 |
| SWE-rebench (Badertdinov et al., 2025) | ✓ | ✓ | Diff | 1 | 3468 | 21,324 |
| SWE-smith (Yang et al., 2025) | ✗ | ✓ | Diff | 1 | 128 | 50,000 |
| SWE-Synth (Pham et al., 2025) | ✗ | ✓ | Diff | 1 | 7 | 9,459 |
| *Natural code-change corpora:* | | | | | | |
| The Stack (Kocetkov et al., 2023) | ✓ | ✗ | None | 30 | – | 317M |
| commitpackft (Muennighoff et al., 2024a) | ✓ | ✗ | Before/After | 277 | – | 742,273 |
| commitbench (Schall et al., 2024) | ✓ | ✗ | Diff | 6 | 72,000 | 1,165,213 |
| CodeReview (Li et al., 2022) | ✓ | ✗ | Diff | 9 | 1,161 | 534,000 |
| **Clean-PR-full (ours)** | ✓ | ✓ | Search/Replace | **12** | **52,338** | **3,050,939** |
| **Clean-PR-train (ours)** | ✓ | ✓ | Search/Replace | **12** | **45,267** | **2,015,708** |

produce the post-PR code exactly. In particular, we utilise Search/Replace edit blocks rather than standard diffs to align with widely adopted code-editing pipelines (Xia et al., 2025; Wang et al., 2025), avoiding the fragility of diffs where valid application hinges on precise line-number prediction (Table 14). **Third, Downstream Sampling:** This process yields a large-scale corpus of approximately **3M PRs** spanning **12** programming languages (46.4B tokens), from which we select a **17.7B** token subset for mid-training by applying modification scope constraints and balanced repository sampling. To the best of our knowledge, this is the **largest publicly released PR-derived dataset** explicitly constructed and validated for repo-level mid-training.

However, mid-training alone does not fully address the challenge of robust localisation and navigation within massive repositories. To bridge this gap, we implement two targeted strategies. First, we align the model with the inference protocol via **Agentless-aligned SFT**, training it to explicitly decompose the problem into (i) file selection, (ii) line-level navigation, and (iii) executable Search/Replace patch generation. Second, to prevent *over-editing* (Zeng et al., 2025) caused by the model's failure to reject irrelevant files within noisy retrieval results, we introduce an **Error-Driven Augmentation** strategy. By injecting distractor files and regions mined from incorrect predictions, we teach the model to reject irrelevant context. This recipe yields consistent improvements on SWE-bench, enhancing both intermediate localisation metrics and end-to-end repair success.

Our contributions can be summarised as follows:

- **Clean-PR Framework:** We propose a data-centric mid-training paradigm transforming noisy GitHub PRs into rigorous training signal. By implementing strict

*Table 2.* Statistics of noise categories applied to the raw dataset. Categories are not mutually exclusive: a single PR may match multiple patterns (e.g., a bot-created PR that is also unmerged).

| Noise Category | Count | Ratio (%) |
|---|---|---|
| Non-Core Source Changes | 6,245,784 | 38.06 |
| Suspected Robot Activity | 4,112,501 | 25.10 |
| Unmerged / Not Approved | 4,016,574 | 24.50 |
| Empty Base File or Diff | 2,241,453 | 13.66 |
| Patch Validation Failure | 675,825 | 4.10 |
| **Clean (w/o Noise)** | **3,050,939** | **18.59** |
| **Total (Raw PR)** | **16,408,886** | **100.00** |

noise filtering and verifying targets through patch application, we bridge the gap between open-source noise and precise repository editing.
- **Largest Verified PR Corpus:** We plan to release the verifiable PR dataset to date (2M PRs). Unlike standard diffs, it utilises a robust Search/Replace format augmented with issue contexts, explicitly constructed to support high-fidelity repository editing tasks.
- **Agentless-Aligned Training:** We propose an SFT strategy with error-driven augmentation to align the model with a simplified Agentless workflow, enabling it to discriminate against distracting context and navigate code without relying on complex agentic loops.
- **Comprehensive Evaluation:** Applying our method to the Qwen2.5-Coder-32B base, we achieve absolute gains of 13.6% on SWE-bench Lite and 12.3% on Verified over baselines. Additional evaluations on 7B models, multilingual repair, and OpenHands show that robust engineering capabilities can be encoded into model weights and transfer across inference paradigms.

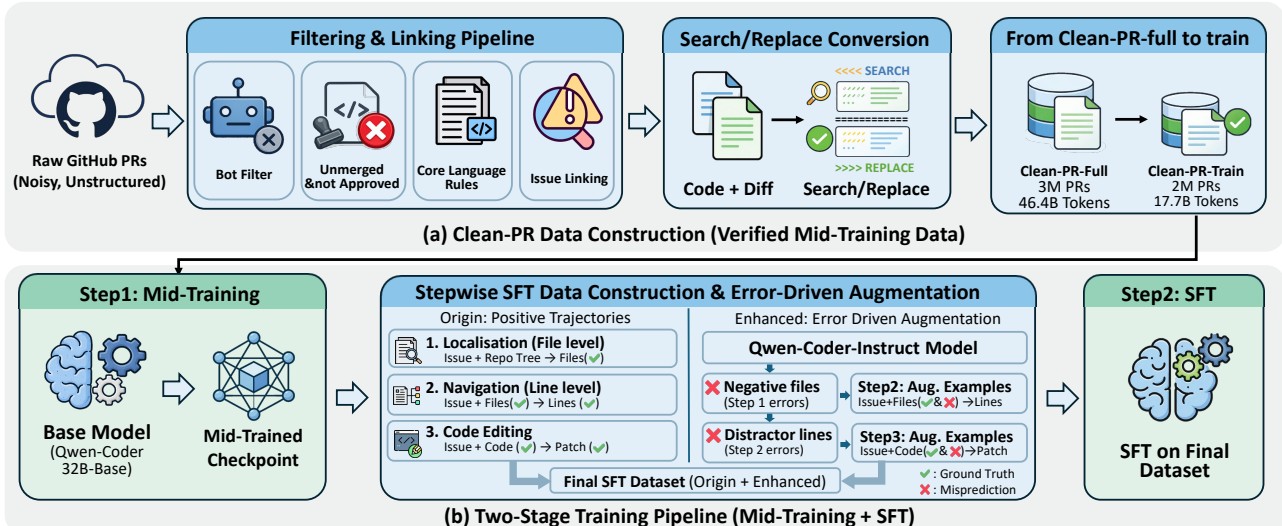

Figure 1. **Overview of the Clean-PR Framework. (a) Data Construction:** Raw GitHub PRs undergo a rigorous filtering pipeline (bot detection, core language enforcement) and intent augmentation via linked Issues. The valid diffs are then converted into minimal unique **Search/Replace** blocks, verified through round-trip patch application to ensure correctness. **(b) Two-Stage Training Pipeline:** The base model first undergoes **Mid-Training** on the verifiable Clean-PR corpus to encode repository-level editing priors. This is followed by an **Agentless-Aligned Stepwise SFT**, where the model is fine-tuned on decomposed tasks (Localisation → Navigation → Editing) with **Error-Driven Augmentation** to robustly handle distracting repository contexts.

## 2. Data Construction

Our objective is to encode repository-level editing capabilities directly into model weights, minimising reliance on heavy inference-time scaffolding. While open-source pull requests (PRs) offer a massive source of developer intent coupled with code changes, raw PR traces are often dominated by noise. To bridge the gap between noisy wild data and the rigorous requirements of repository editing, we construct our data in two stages: (1) **Clean-PR**, a verified mid-training corpus derived from millions of filtered and reconstructed PRs, and (2) an **Agentless-Aligned SFT** dataset designed to bridge the gap between pure editing and the multi-stage localisation-then-editing workflow required by benchmarks like SWE-bench.

### 2.1. Clean-PR: Verified Mid-Training Data

We implement a rigorous pipeline to transform raw GitHub activity into a verifiable training signal. We illustrate a concrete example of a constructed data instance in Figure 4 and provide detailed processing specifications in Appendix A.

**Data Collection.** To construct a comprehensive corpus for repository-level editing, we undertook a crawl of publicly available GitHub pull requests. At the time of this work, our collection comprises approximately **8.6 TB** of raw data, spanning 274k repositories and 16.4 million PRs.

**Data Filtering.** To assess the suitability of raw GitHub data for training, we conducted a preliminary study on the noise distribution within our initial collection. Specifically, we defined a noise taxonomy by synthesizing heuristics from (Lozhkov et al., 2024; Kocetkov et al., 2023) with manual inspection. As detailed in Table 2, the raw stream is heavily polluted: 38.06% of PRs lack core source changes, 25.10% stem from bot activity, and 24.50% remain unmerged. Motivated by these findings, we apply a rigorous filtering protocol to aggressively prune these non-learning signals, retaining only **18.59%** of the original data as high-density training signals. Crucially, to prevent data contamination, we explicitly exclude all repositories present in the SWE-bench evaluation sets from our final corpus.

- **PR Validity:** Raw GitHub data contains substantial noise. We discard PRs that are unmerged, closed without merging, or created solely by automated accounts (bots). We further exclude PRs that exclusively change documentation, which are unsuitable for learning focused semantic edits.
- **Language Alignment:** Repository-level training must emphasise semantic code editing rather than configuration churn. We enforce a *Core Extension Rule*, retaining a PR only if it modifies at least one core source file corresponding to our 12 target languages, with specific definitions provided in Table 15. PRs dominated by configuration or auto-generated files are removed to ensure the model learns meaningful logic changes.

**Search/Replace format Reconstruction.** Standard unified diffs are brittle for LLM generation due to their reliance on fragile line numbers. We instead utilise the

*Table 3.* Clean-PR mid-training corpus statistics (stage-wise).

| PR Cleaning Stage | #PRs | Tokens |
|---|---|---|
| Raw PR snapshot (with code) | 16,408,886 | – |
| Filtering + S/R format (**Clean-PR-full**) | 3,050,939 | 46.4B |
| Core-file constraint ($\leq 5$ core files) | 2,751,937 | 29.7B |
| After format/length constraints | 2,585,778 | 22.0B |
| **Clean-PR-train (repo-sampling)** | **2,015,708** | **17.7B** |

*Table 4.* Detailed statistical comparison between the full verified corpus and the final training set.

| Metric | Clean-PR-full | Clean-PR-train |
|---|---|---|
| Total Instances | 3,050,939 | 2,015,708 |
| Total Repositories | 52,338 | 45,267 |
| Total Tokens | 46.4B | 17.7B |
| Avg. Description Len (words) | 50.0 | 59.5 |
| Avg. Modified Files | 3.0 | 1.7 |
| Avg. Code Lines | 1,562.3 | 1,077.8 |
| Avg. Search/Replace Blocks | 9.0 | 4.3 |
| Avg. Search/Replace Lines | 130.3 | 58.7 |
| Avg. Comments | 2.2 | 2.1 |

**Search/Replace** format (Xia et al., 2025), which locates edits via unique context matching. To ensure these targets are grounded in a valid repository state, we perform a rigorous round-trip verification. The detailed algorithmic procedure is outlined in Algorithm 1. First, we reconstruct the exact "after" state of the repository by applying the raw PR patch to the "before" snapshot. We then algorithmically derive minimal edit spans and select unique anchor contexts to form Search/Replace blocks. Finally, we *verify* these blocks by applying them back to the "before" state; any example where the re-application does not bit-wise match the ground-truth "after" state is discarded. This guarantees that every training example corresponds to a **minimal unique search block**, defined as the shortest contiguous span of context needed to uniquely identify the edit location within a file, thereby filtering out noisy diffs caused by formatting drift.

**Issue-Augmented Intent.** Raw PR descriptions frequently lack self-contained context, as developers customarily reference an external Issue identifier (e.g., "Fixes #123") rather than restating the detailed bug report or feature requirement. To recover this missing problem definition, we implement an issue augmentation pipeline: we retrieve and concatenate the titles and descriptions of all linked issues into the PR context. By supplementing brief developer summaries with original user reports, we align the training signal more closely with real-world software engineering workflows, where the initial prompt is typically a detailed bug report rather than a known solution.

**From Clean-PR-full to Clean-PR-train.** As detailed in Table 3, our initial filtering yields the **Clean-PR-full** corpus,

comprising 3.05 million verified instances (46.4B tokens). To construct the final **Clean-PR-train** dataset, we apply three targeted refinements. First, to focus the model on self-contained, learnable units of work, we implement **complexity control** by restricting the dataset to PRs modifying at most five core files; as shown in Table 4, this reduces the average modified files from 3.0 to 1.7. Second, we apply **context windowing** to files exceeding 100k tokens, preserving the editing signal by centring the input window around verified Search/Replace blocks. Finally, to mitigate distribution skew from dominant projects, we enforce **repository-level sampling**: for any repository contributing more than 2,000 PRs, we randomly sample exactly 2,000 instances, while retaining all PRs from smaller repositories, resulting in a balanced corpus of **2 million** instances. The detailed language distribution is provided in Appendix A.8, with Python PRs accounting for 19.3% of the training set.

## 2.2. Agentless-Aligned Stepwise SFT

Mid-training equips the model with the fundamental capability to edit code given a context. However, resolving repository-scale issues necessitates a structured **decompose-and-solve** workflow: locating files, identifying regions, and then editing a paradigm established as robust by recent systems (Xia et al., 2025; Yang et al., 2024; Jiang et al., 2025). To bridge the gap between our raw editing capability and this structured requirement, we construct a high-quality SFT dataset aligned with a Simplified Agentless workflow. We favour this generative approach over complex Agentic frameworks as it leverages the strong reasoning priors of our base model without the overhead of multi-turn dialogue or external tool use. We achieve this by leveraging verified trajectories from SWE-rebench (Badertdinov et al., 2025) and SWE-Gym (Pan et al., 2025), effectively "baking" fine-grained navigation intelligence directly into the model weights to reduce reliance on external scaffolding.

**Task Decomposition and Filtering.** We derive training samples by decomposing each ground-truth repair into three distinct supervised tasks, rigorously filtering for tractability:

- **File localisation** (Issue + Repo Tree → Filepath): Navigating real-world codebases is a needle-in-a-haystack challenge; for instance, repositories in SWE-bench contain an average of 3,010 files, yet a typical issue requires modifying only 1.7 files (Jimenez et al., 2024). Therefore, effectively narrowing the search space is a prerequisite for tractable editing. In this step, the model learns to identify relevant source files solely from the issue description and repository structure. To focus the training signal on functional repairs, we filter the target labels to exclude non-code artefacts (e.g., `.md`, `.txt`), ensuring the model learns to prioritise

*Table 5.* Statistics of the Agentless-Aligned SFT dataset. *Origin* refers to positive samples derived from source benchmarks (SWE-bench-Live, SWE-rebench, SWE-Gym). *Enhanced* refers to hard negative samples generated via error-driven augmentation.

| Data Source | Step 1 | Step 2 | Step 3 | Total |
|---|---|---|---|---|
| Origin | 18,891 | 17,763 | 16,564 | 53,218 |
| Error-Augmented | – | 12,989 | 8,875 | 21,864 |
| **Total SFT Data** | **18,891** | **30,752** | **25,439** | **75,082** |

semantic code changes over auxiliary file updates.

- **Fine-grained Navigation** (Issue + File Content → Relevant Context): Since source files often span thousands of lines, operating on full-file context is inefficient and prone to distraction. We utilise Abstract Syntax Tree (AST) parsing to map ground-truth edits to their enclosing function or class definitions. This trains the model to identify the precise logical scope of the bug, rather than arbitrary line numbers, ensuring robustness against minor formatting shifts.
- **Patch Generation** (Localised Context → Search /Replace Patch): Given the identified code region, the model generates the final fix. We enforce that the target output forms a **minimal unique search block**, which prevents ambiguous application errors while minimising token usage compared to verbose context blocks.

**Error-Driven Augmentation.** A critical challenge in repository-level coding is robustness to distracting context. Standard SFT trains on the "happy path" of perfect localisation. However, in real-world inference, retrieval is imperfect; without training to handle noise, LLMs are prone to *over-editing*: mistakenly modifying irrelevant files or unchanged regions because they are provided in context (Zeng et al., 2025). To mitigate this, we employ an intermediate model (Qwen-2.5-Coder-32B-Instruct) to generate realistic noise. For Fine-grained Navigation, we combine ground-truth files ($F_{gt}$) with erroneous hard negatives ($F_{neg}$) predicted by the intermediate model to define the training mapping: *Issue* + ($F_{gt} \cup F_{neg}$) → *Relevant Context*. Here, the model learns to extract *Relevant Context* from $F_{gt}$ while returning "No changes needed" for $F_{neg}$.

Similarly, we harden the **Patch Generation (Step 3)** against noisy context. We utilise the intermediate model to retrieve distractor code regions ($C_{noise}$) that are semantically similar to the bug location but require no editing. By combining these with the correct content ($C_{relevant}$) to form the input *Localised Context*, the training objective is defined as: *Issue* + ($C_{relevant} \cup C_{noise}$) → *Search/Replace*. This data-centric approach ensures the model discriminates based on the specific issue intent rather than merely operating on the assumption of perfect retrieval.

Table 5 summarises the final composition of the SFT dataset. The *Origin* set represents the clean trajectories derived from the source benchmarks, while the *Error-Augmented* set comprises the negative samples generated via our error-driven augmentation pipeline to improve robustness against noise.

## 3. Experiments

### 3.1. Experiment Setup

**Training Configurations.** We initialise our mid-training from Qwen2.5-Coder-32B-Base (Hui et al., 2024) and conduct all experiments on a cluster of 32 NVIDIA H200 GPUs with a context window of 32,768 tokens. For ablation analysis, we define a "Python Only" setting trained exclusively on the Python subset of Clean-PR-train, contrasting it with the full multi-language corpus. In terms of computational cost, this "Python Only" mid-training requires approximately 60 wall-clock hours, significantly less than the 259 hours for the full "All Languages" setting, while the final stepwise SFT stage completes in 38 hours. Comprehensive hyperparameter settings are provided in Appendix C.

**Benchmarks and Metrics.** We evaluate Clean-PR on SWE-bench Lite (300 instances) and Verified (500 instances) (Jimenez et al., 2024). We report four key metrics: (1) **Pass@1**, the primary metric for issue resolution; (2) **Valid Patch Rate**, measuring the percentage of generated patches that are applied successfully; and (3) intermediate retrieval metrics including **File localisation Accuracy** and **Line Accuracy**, which precisely quantify the model's ability to locate correct files and edit spans, respectively.

**Inference Scaffold.** Crucially, we adopt a **Simplified Agentless** scaffolding (Xia et al., 2025) (detailed in Section 2.2) rather than a complex Agent-based framework. We choose this deterministic protocol for two reasons: (1) **Lightweight Evaluation:** The Agentless workflow encapsulates the standard problem-solving stages (localisation, patch generation) found in most agentic frameworks but executes them in a linear, efficient manner. This avoids the heavy computational overhead of iterative execution loops, enabling rapid and scalable benchmarking. (2) **Isolation of Gains:** This streamlined workflow allows us to **more clearly and reliably** isolate and measure the intrinsic editing capabilities acquired from our data pipeline, disentangling our contribution from the variance introduced by complex planning loops or prompt engineering strategies.

**Internal Baselines: Data Strategy Ablation.** We compare Clean-PR against three controlled settings based on Qwen2.5-Coder-32B. First, we use its official Instruct model as a zero-shot baseline to represent generalist capabilities. Second, we evaluate Base + SFT (without mid-training) to

*Table 6.* Main results on SWE-bench Lite and Verified. The table is organised by benchmark split. All reported metrics are percentages. **File Acc.** denotes file-level localisation recall. **Line Acc.** denotes fine-grained navigation accuracy. **Valid** indicates the percentage of generated patches that are syntactically valid and can be applied successfully. **Pass@1** reports the percentage of issues resolved.

| Base Model | Mid-Train Setting | SFT | Valid Patch | File Acc. | Line Acc. | Pass@1 |
|---|---|---|---|---|---|---|
| *SWE-Bench Lite (300 instances)* | | | | | | |
| Qwen-Coder-32B-Instruct | None | ✗ | 77.0 | 74.7 | 38.3 | 10.7 |
| Qwen-Coder-32B-Base | None | ✓ | 84.0 | 78.3 | 46.7 | 11.3 |
| Qwen-Coder-32B-Base | StarCoder2-Style Dataset (All, 17.4B) | ✓ | 89.7 | 84.3 | 47.0 | 15.7 |
| Qwen-Coder-32B-Base | **Clean-PR-train (Ours, Python, 3.8B)** | ✓ | 95.7 | 86.3 | 54.0 | 22.3 |
| | **Clean-PR-train (Ours, All, 17.7B)** | ✓ | **96.3** | **87.3** | **55.7** | **24.3** |
| *SWE-Bench Verified (500 instances)* | | | | | | |
| Qwen-Coder-32B-Instruct | None | ✗ | 77.6 | 70.6 | 42.3 | 18.3 |
| Qwen-Coder-32B-Base | None | ✓ | 81.8 | 74.3 | 46.6 | 17.6 |
| Qwen-Coder-32B-Base | StarCoder2-Style Data (All, 17.4B) | ✓ | 82.4 | 77.7 | 48.4 | 20.4 |
| Qwen-Coder-32B-Base | **Clean-PR-train (Ours, Python, 3.8B)** | ✓ | 94.4 | 78.5 | 51.6 | 27.8 |
| | **Clean-PR-train (Ours, All, 17.7B)** | ✓ | **95.2** | **80.7** | **52.2** | **30.6** |

*Table 7.* Comparison with open-source methods (pass@1); results are copied from the original paper. All metrics are percentages.

| Method | Framework | Params | Lite | Verified |
|---|---|---|---|---|
| SWE-Gym | OpenHands | 32B | 15.3 | 20.6 |
| Lingma-SWE | SWESynInfer | **72B** | 22.0 | 30.2 |
| SWE-Fixer | SWE-Fixer | **72B** | 22.0 | 30.2 |
| **Clean-PR** | Agentless | 32B | **24.3** | **30.6** |

establish a lower bound for instruction tuning. Third, and most critically, we implement a StarCoder2-style Baseline (Lozhkov et al., 2024). This baseline represents the prevailing standard for training on GitHub data (Appendix B) but differs from Clean-PR in three fundamental aspects: (1) **Format:** It utilises the noisy *Unified Diff* format rather than our verifiable Search/Replace blocks; (2) **Filtering:** It retains non-code artefacts (e.g., JSON, YAML config files), whereas we enforce strict Core Language filtering; and (3) **Issue-Augmented Context:** Unlike standard practice which processes PRs and Issues in isolation, Clean-PR integrates linked Issue descriptions into the training sequence. This forces the model to learn the alignment between natural language intent and code implementation.

**External Baselines: Open-Source SOTA.** We further benchmark Clean-PR against representative high-performing open systems to contextualise our efficiency. We include SWE-Gym (Pan et al., 2025), which shares our model size (32B) but employs the complex OpenHands agentic framework with iterative planning. Additionally, we compare against substantially larger models, specifically Lingma-SWE (Ma et al., 2025) and SWE-Fixer (Xie et al., 2025). Both utilise 72B-parameter base models.

## 3.2. Main Results

Table 6 presents the evaluation on SWE-bench Lite and Verified. We analyse the results across three key dimensions: the effectiveness of mid-training, the impact of Clean-PR pipeline, and the benefits of multi-language training.

**Effectiveness of Mid-Training.** The results highlight the benefit of incorporating a dedicated repository-level mid-training stage. The Base + SFT model, despite being fine-tuned on the stepwise dataset, achieves only 11.3% on Lite and 17.6% on Verified. In contrast, introducing any form of repository-level mid-training, even the noisy StarCoder2-style baseline, yields immediate gains. This step boosts Lite performance to 15.7% (+4.4%) and Verified to 20.4% (+2.8%), confirming that pre-encoding repository structures and editing patterns into the model weights is a prerequisite for effective downstream performance.

**Superiority of Clean-PR Data Construction.** Comparing StarCoder2-style (17.4B) with Clean-PR-train (17.7B) shows a clear advantage, with our method reaching 24.3% on Lite and 30.6% on Verified. We hypothesize these gains stem from our strict data structuring. First, the verified **Search/Replace** format correlates with improved *Valid Patch* rates (89.7% → 96.3%). Unlike line-number-based Diffs, S/R requires explicit context matching, which likely grounds the model's edits and reduces format application errors. Second, the rise in *Line Localisation* (47.0% → 55.7%) suggests that training on unique search blocks encourages the model to generate more precise, unambiguous navigation cues compared to raw noisy diffs.

**Benefits of Multi-Language Training.** We find that multi-language training yields additional performance gains. Although the Python Only model (3.8B tokens) performs impressively, surpassing the 17.4B token StarCoder baseline

*Table 8.* Effect of data source and edit format on SWE-Bench performance. We bold our default setting for comparison.

| Data source | Edit Format | Description | Lite | Verified |
|---|---|---|---|---|
| StarCoder-style | Diff | PR Desc Only | 15.7 | 20.4 |
| Clean-PR-train (Python) | **Search/Replace** | **Linked Issue** | **22.3** | **27.8** |
| | Diff | Linked Issue | 19.1 | 24.4 |
| | Search/Replace | PR Desc Only | 20.4 | 25.7 |

*Table 9.* Ablation study of SFT data strategies on SWE-Bench Lite and Verified.

| Lang | SFT Strategy | Lite | | | Verified | | |
|---|---|---|---|---|---|---|---|
| | | File | Line | Pass@1 | File | Line | Pass@1 |
| Python | Standard | 86.7 | 51.3 | 18.7 | 78.3 | 49.4 | 24.3 |
| | **Error Aug.** | **86.3** | **54.0** | **22.3** | **78.5** | **51.6** | **27.8** |
| All | Standard | 87.0 | 53.0 | 21.8 | 80.3 | 50.0 | 27.4 |
| | **Error Aug.** | **87.3** | **55.7** | **24.3** | **80.7** | **52.2** | **30.6** |

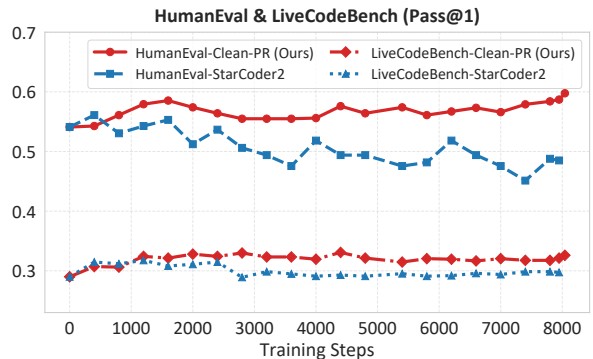

*Figure 2.* Generalisation dynamics during mid-training.

with 22.3% on Lite, scaling to All Languages (17.7B tokens) yields the best overall results (24.3% on Lite and 30.6% on Verified). This suggests that exposure to diverse syntactical structures, such as those from Java, C++, and Go, enhances the abstract reasoning capabilities of the model in issue solving/software engineering.

**Comparison with Recent Open-Source methods.** Table 7 benchmarks Clean-PR against representative open-source methods. Using the same 32B base, Clean-PR significantly outperforms SWE-Gym (30.6% vs 20.6% on Verified) which relies on complex agent scaffolding. Remarkably, despite having half the parameters, our model surpasses the 72B baselines (Lingma-SWE and SWE-Fixer) on both Lite and Verified. This confirms that rigorous mid-training bridges the scaling gap, enabling SOTA performance under a lightweight workflow without expensive iterative loops.

### 3.3. Ablation Studies

**Contributions of Linked Issue and Edit Format.** To rigorously disentangle the individual contributions of our data construction pipeline, we conduct an ablation study on the Python subset of our mid-training data, as detailed in Table 8, with two main findings. **First, the edit format is critical.** Replacing our verified Search/Replace blocks with standard Unified Diffs results in a performance drop (e.g., from 27.8% down to 24.4% on Verified). This confirms that the deterministic, context-rich nature of Search/Replace blocks provides a far more robust training signal than brittle diff lines. **Second, augmenting context provides a realistic problem definition.** Relying solely on raw PR descriptions degrades performance to 25.7% on Verified. By incorporating linked issue descriptions, we provide the model with the original problem definition rather than just the solution summary, yielding a clear gain. Notably, the combination of both strategies achieves the best performance, significantly outperforming the StarCoder-style baseline which lacks both rigorous formatting and intent augmentation.

**Impact of Error-Driven Augmentation.** To validate the effectiveness of our augmentation strategy, we compare the performance of models fine-tuned on the standard SFT dataset versus the version augmented with hard negatives and distractor regions. As shown in Table 9, this strategy yields consistent gains across all settings. For our

best-performing "All Languages" model, the augmentation boosts the Pass@1 rate from 21.8% to **24.3%** on SWE-bench Lite and from 27.4% to **30.6%** on SWE-bench Verified. Crucially, we observe simultaneous improvements in Line accuracy, which confirms that explicitly training the model to discriminate against distracting context and reject irrelevant files significantly enhances its robustness and precision in real-world repository navigation.

**Generalisation Capability and Catastrophic Forgetting.** A critical challenge in repository-specific adaptation is avoiding the loss of general programming capabilities ("catastrophic forgetting") (van de Ven et al., 2025). We visualise the training dynamics in Figure 2. The StarCoder2-style baseline, trained on standard diffs, exhibits a clear degradation trend: HumanEval performance drops from 54.1% to 47.6% (-6.5%) as training progresses. This suggests that raw diffs may hinder the model's core reasoning due to fragile line numbers and unverified context. In stark contrast, Clean-PR demonstrates robust positive transfer. By learning from **verified Search/Replace** blocks, the model not only preserves its pre-trained capabilities but actively sharpens them, reaching 59.8% on HumanEval (+5.7%) and boosting LiveCodeBench from 29.0% to 32.6%. This suggests that the precise context matching required by our objective transfers effectively to general code generation, proving that repository-level adaptation need not come at the cost of fundamental coding skills.

**Scaling Inference with Best-of-N.** We explore the upper bound of our model's capability by evaluating Pass@k per-

*Table 10.* Scale generalisation on Qwen2.5-Coder-7B. The table is organised by benchmark split. All reported metrics are percentages. **File Acc.** denotes file-level localisation recall. **Line Acc.** denotes fine-grained navigation accuracy. **Valid** indicates the percentage of generated patches that are syntactically valid and can be applied successfully. **Pass@1** reports the percentage of issues resolved.

| Base Model | Mid-Train | SFT | Valid Patch | File Acc. | Line Acc. | Pass@1 |
|---|---|---|---|---|---|---|
| *SWE-Bench Lite (300 instances)* | | | | | | |
| SWE-Gym (Qwen2.5-Coder-7B-Instruct) | None | ✓ | – | – | – | 10.0 |
| Lingma-SWE-GPT-7B | External | – | – | – | – | 12.0 |
| Qwen-Coder-7B-Instruct | None | ✗ | 6.3 | 62.7 | 24.4 | 1.3 |
| Qwen-Coder-7B-Base | None | ✓ | 81.2 | 66.3 | 32.0 | 10.3 |
| Qwen-Coder-7B-Base | **Clean-PR** | ✓ | **90.7** | **82.0** | **42.7** | **14.5** |
| *SWE-Bench Verified (500 instances)* | | | | | | |
| SWE-Gym (Qwen2.5-Coder-7B-Instruct) | None | ✓ | – | – | – | 10.6 |
| Lingma-SWE-GPT-7B | External | – | – | – | – | 18.2 |
| Qwen-Coder-7B-Instruct | None | ✗ | 5.7 | 57.6 | 22.6 | 2.4 |
| Qwen-Coder-7B-Base | None | ✓ | 80.1 | 61.3 | 29.5 | 14.2 |
| Qwen-Coder-7B-Base | **Clean-PR** | ✓ | **90.8** | **77.0** | **43.2** | **20.4** |

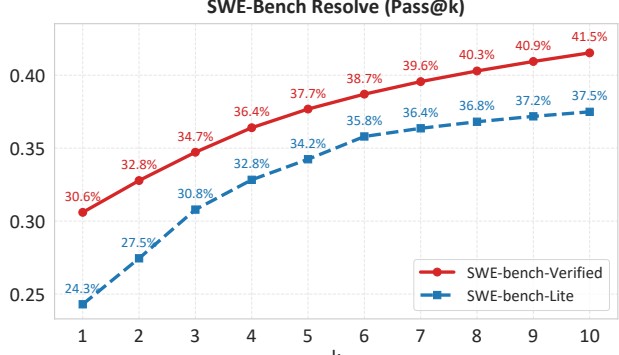

*Figure 3.* Pass@k performance on SWE-bench Lite and Verified. We report the resolution rates of our model (Clean-PR, mid-trained on All Languages) as the number of samples $k$ scales.

formance, where the model generates $k$ candidate patches for each issue. As illustrated in Figure 3, Clean-PR benefits substantially from increased sampling. On SWE-bench Verified, the performance improves monotonically from 30.6% at $k = 1$ to 41.5% at $k = 10$. Similarly, on SWE-bench Lite, the resolution rate rises from 24.3% to 37.5%. This gap between Pass@1 and Pass@10 suggests that while the model has the intrinsic reasoning capability to solve a large portion of issues, the standard likelihood-based ranking is not always perfectly aligned with functional correctness. These results indicate that integrating a lightweight re-ranking mechanism or a verifier could further unlock the model's potential without requiring expensive agentic training.

### 3.4. Additional Generalisation Studies

**Scale Generalisation.** We further investigate whether the benefits of Clean-PR generalise across model scales beyond the default 32B setting. Specifically, we apply the same mid-training and SFT pipeline to Qwen2.5-Coder-7B

*Table 11.* Multilingual evaluation on Multi-SWE-bench Flash. All metrics are percentages.

| Base Model | Mid-Train | SFT | Valid | File | Line | Pass@1 |
|---|---|---|---|---|---|---|
| Qwen-Coder-32B-Instruct | None | ✗ | 71.7 | 40.0 | 15.2 | 6.7 |
| Qwen-Coder-32B-Base | None | ✓ | 73.0 | 42.3 | 17.7 | 7.0 |
| Qwen-Coder-32B-Base | StarCoder2 | ✓ | 76.3 | 46.0 | 20.3 | 8.7 |
| Qwen-Coder-32B-Base | **Clean-PR** | ✓ | **81.7** | **51.3** | **24.0** | **12.3** |

and compare against both internal and external 7B baselines (SWE-Gym fine-tuned on Qwen2.5-Coder-7B-Instruct, and Lingma-SWE-GPT-7B). Table 10 shows that the same recipe consistently transfers to Qwen2.5-Coder-7B, improving Pass@1 from 10.3% to 14.5% on Lite and from 14.2% to 20.4% on Verified over the SFT-only base. Notably, the localisation gains are larger than for the 32B model, suggesting that high-quality PR supervision is especially useful when model capacity is limited.

**Multilingual Evaluation.** The main SWE-bench splits are Python-only, whereas Clean-PR-train spans 12 languages. Table 11 therefore evaluates multilingual issue-solving on Multi-SWE-bench Flash (Zan et al., 2025), which contains 300 instances across C, C++, Go, Java, JavaScript, Rust, and TypeScript. For multilingual SFT, we use Multi-SWE-bench-RL, yielding 9,331 samples from 69 repositories with no overlap with SWE-bench Lite/Verified or Multi-SWE-bench. Clean-PR reaches 12.3% Pass@1, outperforming both the instruct baseline and the StarCoder2-style baseline, with consistent gains in valid-patch and localisation.

**Agentic Evaluation.** Finally, we test whether the internalised editing capability transfers back into a multi-turn agent. Following SWE-Gym, we train an agent variant with 491 successful OpenHands trajectories (Wang et al., 2025) and evaluate on OpenHands v0.28 using CodeActAgent with a 100-turn limit. As shown in Table 12, Clean-PR

*Table 12.* Agentic evaluation with OpenHands CodeActAgent.

| Model | Empty ↓ | Loop ↓ | Turns | Resolve ↑ |
|---|---|---|---|---|
| *SWE-Bench Lite (300 instances)* | | | | |
| SWE-Gym | – | – | – | 15.3 |
| Qwen-Coder-32B-Instruct | 42.3 | 42.8 | 29.3 | 2.8 |
| Qwen-Coder-32B-Base+SFT | 21.0 | 28.7 | 37.8 | 16.1 |
| Qwen-Coder-32B-Base+**Clean-PR**+SFT | **16.3** | **26.7** | 42.5 | **20.7** |
| *SWE-Bench Verified (500 instances)* | | | | |
| SWE-Gym | – | – | – | 20.6 |
| Qwen-Coder-32B-Instruct | 28.4 | 37.8 | 27.4 | 6.3 |
| Qwen-Coder-32B-Base+SFT | 16.6 | **23.7** | 34.1 | 19.5 |
| Qwen-Coder-32B-Base+**Clean-PR**+SFT | **14.3** | 24.7 | 38.4 | **24.7** |

improves over SFT-only by 4.6 points on Lite and 5.2 points on Verified, while also reducing empty-patch rates. Since OpenHands uses a different action space and failure modes from our Simplified Agentless scaffold, this suggests that Clean-PR provides a stronger editing prior for agents rather than merely matching a fixed inference template.

## 4. Related Work

**Inference Paradigms and System Complexity.** The pursuit of automated repository-level engineering has spurred a diverse ecosystem of inference frameworks. Early dominant approaches relied on **Agentic frameworks**, where models function as autonomous agents interacting with an environment via tools (e.g., shell, file editors). Systems like SWE-agent (Yang et al., 2024), OpenHands (Wang et al., 2025), and AutoCodeRover (Zhang et al., 2024) employ iterative reasoning loops to navigate codebases, though they often suffer from error propagation in long trajectories. In response, **Agentless paradigms** (Xia et al., 2025) emerged as a streamlined alternative, decomposing the problem into static retrieval, precise localisation, recently enhanced by code-structure signals like call graphs (Jiang et al., 2025), and constrained patch synthesis. However, the recent trend has shifted towards overcoming model limitations through increasing system complexity and **Test-Time Scaling**. This includes the development of reinforcement learning environments (Pan et al., 2025; Jain et al., 2025) for policy optimisation, contamination-aware evaluation protocols (Badertdinov et al., 2025), and compute-intensive search strategies that sample and rerank candidate trajectories (Antoniades et al., 2025). Recent systems such as SWE-Swiss-32B (He et al., 2025) further combine multi-task SFT, reinforcement learning, and self-consistency decoding, which are complementary to our focus on data-centric mid-training. Furthermore, benchmarks have evolved to challenge these systems with dynamic issue streams (Zhang et al., 2025), multilingual repositories (Zan et al., 2025; Rashid et al., 2025), and long-context understanding (Rando et al., 2025). While these engineering advancements drive higher scores, they often obscure the intrinsic capability of the underlying model. Our work focuses on internalising these repository-editing skills directly into model weights, reducing the dependency on heavy inference scaffolding.

**Evolution of Code Training: From Files to PRs.** The efficacy of code models is fundamentally constrained by the granularity of their training data, which has evolved through three distinct levels. **1) File-Level:** Foundational models like StarCoder (Lozhkov et al., 2024) and Qwen-Coder(Hui et al., 2024) are pre-trained on massive file collections such as The Stack (Kocetkov et al., 2023). While this provides vast syntactic knowledge, it treats code as static snapshots, lacking the temporal context of software evolution. **2) Commit-Level:** To capture editing dynamics, recent work leverages version-control diffs and commit metadata, ranging from instruction tuning on commits (e.g., Commit-PackFT (Muennighoff et al., 2024b), Commitbench (Schall et al., 2024)) to commit- and edit-centric pretraining objectives (e.g., CoditT5 (Zhang et al., 2023), CommitBART (Liu et al., 2024), Coeditor (Wei et al., 2024a)). However, commits/diffs still provide weak or fragmented intent signals (Tian et al., 2022), and rarely capture the full multi-file context and discussion that drive real engineering work. **3) PR-Level:** Pull Requests represent the ideal training signal, offering a comprehensive view that couples high-level human intent with extensive, multi-file code modifications (Gousios et al., 2014; Tsay et al., 2014). Despite their potential, leveraging PRs is notoriously difficult due to the "noise-validity gap" in mining GitHub at scale, further exacerbated by PR-specific artefacts such as bot-generated activity (Golzadeh et al., 2020; Wessel & Steinmacher, 2020), and the prevalence of unmerged or abandoned contributions that lack verifiable quality assurance. Consequently, prior PR-centric works have been limited to auxiliary tasks like code review (Li et al., 2022) or synthetic data generation (Wei et al., 2024b; Yang et al., 2025), rather than direct training for editing. We bridge this gap by proposing **Clean-PR**, a scalable pipeline that rigorously filters and verifies PRs to construct a massive, deterministic corpus, enabling models to learn repository-level editing at scale.

## 5. Conclusion

In this work, we addressed the scarcity of high-quality supervision for repository-level engineering by introducing **Clean-PR**. We transformed noisy GitHub pull requests into a rigorous corpus of **2 million** verifiable **Search/Replace** instances (17.7B tokens). To harness this data, we proposed an **Agentless-aligned stepwise SFT** strategy augmented with error-driven negative sampling. Our experiments demonstrate that this data-centric approach enables a 32B model to achieve highly competitive performance without heavy agent scaffolding, while additional 7B, multilingual, and OpenHands evaluations show that the learned repository-editing capability transfers across scale, language, and inference paradigms.

## Impact Statement

This work introduces Clean-PR to advance automated repository-level software engineering. While our approach holds significant potential to enhance developer productivity and lower the barrier for open-source maintenance, it also introduces challenges regarding the reliability and security of generated code. As models gain the capability to modify complex codebases, there are risks of introducing subtle vulnerabilities or being misused for malicious automation. Furthermore, the use of public repositories necessitates ongoing attention to licensing and attribution. We encourage the community to prioritise the development of robust verification tools and safety guardrails to ensure these agents serve as reliable, human-aligned assistants.

## Acknowledgments

This work was supported in part by the UK Engineering and Physical Sciences Research Council (EPSRC) through a Turing AI Fellowship (grant no. EP/V020579/1, EP/V020579/2) and a New Horizons grant (grant no. EP/X019063/1), and KCL's Impact Acceleration Account (grant no. EP/X525571/1). The work of Xiangxiang Dai was supported by the National Natural Science Foundation of China (625B2163).

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

# Appendix

*Table 13.* **Visualisation of Raw Data Entities.** Note that Issues and PRs are distinct objects. The Issue contains the natural language description of the bug, while the PR contains the metadata, code context, and the diff. We link them during the data construction phase.

---

**DATA INSTANCE 1: Raw Issue #4288 (User Intent)**

**Repo:** `example/repository`  **Issue ID:** 4288  **Author:** `@user_A`
**Created At:** 2024-03-14 10:00:00  **Status:** `Closed`

---

**Title:** `IndexError when batch size is 1 in attention mask`
**Description Body:**
When passing a single sequence to the model (batch size = 1), the attention mask expansion fails.

```
Traceback (most recent call last):
  File "src/models/attention.py", line 45, in expand_mask
    bsz, src_len = mask.size()
ValueError: not enough values to unpack (expected 2, got 1)
```

It seems we strictly expect a batch dimension $> 1$. This works fine when $bsz > 1$ but crashes on single inference.

---

**DATA INSTANCE 2: Raw PR #4290 (Implementation)**

**Repo:** `example/repository`  **PR ID:** 4290  **Author:** `@dev_B`
**Status:** `Merged`  **Base Commit:** `7b3f1a2`  **Head Commit:** `9c4e2d1`

---

**PR Title:** `Fix crash on empty input list`
**PR Description:** `Fixed the unpacking error reported in #4288. Added specific shape check.`

**File Context (Base Code before edit):**  *Path: src/models/attention.py*

```
38  def expand_mask(mask, dtype, tgt_len):
39      """
40      Expands attention_mask to [bsz, 1, tgt_len, src_len].
41      """
42      # CAUTION: This line causes overflow if not careful
43      bsz, src_len = mask.size()
44      tgt_len = tgt_len if tgt_len is not None else src_len
45
46      expanded_mask=mask[:,None,None,:].expand(bsz,1,tgt_len,src_len)
47      return expanded_mask.to(dtype)
```

**Diff Hunk (The Change):**

```
diff --git a/src/models/attention.py b/src/models/attention.py
--- a/src/models/attention.py
+++ b/src/models/attention.py
@@ -43,3 +43,5 @@ def expand_mask(mask, dtype, tgt_len):
-     bsz, src_len = mask.size()
+     # Robust unpacking for single batch
+     shape = mask.shape
+     bsz, src_len = shape[0], shape[-1]
      tgt_len = tgt_len if tgt_len is not None else src_len
```

**Comments:**
[@maintainer]: Verified. The .shape access is safer here. Merging.

*Table 14.* **Why Convert? Diff vs. Search/Replace.** (Left) Raw Git Diffs use line numbers (e.g., `@@ -43,3`), which makes them fragile for training; if the code shifts by one line, the label becomes invalid. (Right) Search/Replace format uses unique context strings to anchor the edit, ensuring robustness.

| Raw Format: Unified Diff | Training Format: Search/Replace |
|---|---|
| ```diff --git a/attention.py b/attention.py --- a/attention.py +++ b/attention.py @@ -43,3 +43,5 @@  <-- FRAGILE -    bsz, src_len = mask.size() +    shape = mask.shape +    bsz, src_len = shape[0], shape[-1]      tgt_len = tgt_len...``` | ```<<<<<<< SEARCH     bsz, src_len = mask.size()     tgt_len = tgt_len if tgt_len =======     shape = mask.shape     bsz, src_len = shape[0], shape[-1]     tgt_len = tgt_len if tgt_len >>>>>>> REPLACE``` |
| **Cons:** Relies on `line 43`. If upstream changes move this to `line 45`, the patch fails. | **Pros:** Matches the text `bsz, src_len...` anywhere in the file. |

*Table 15.* Language definitions used for data filtering. A PR is assigned to a language $L$ if it modifies at least one **Core** file of $L$, and contains only files from the **Allowed** list of $L$.

| Language | Core Extensions | Allowed Extensions (Context & Config) |
|---|---|---|
| Python | .py | .py, .md, .rst, .txt, .yml, .yaml, .toml, .cfg, .ini, .json, .png, .jpg, .jpeg, .svg, .gif, .html, .sh, .bash |
| Java | .java | .java, .xml, .properties, .gradle, .md, .txt, .json, .yml, .yaml, .png, .jpg, .jpeg, .svg, .gif, .html, .css, .js, .sh |
| TypeScript | .ts, .tsx | .ts, .tsx, .js, .jsx, .json, .md, .txt, .yml, .yaml, .png, .jpg, .jpeg, .svg, .gif, .vue, .html, .css, .scss, .sass, .less, .sh, .graphql, .gql |
| Go | .go | .go, .mod, .sum, .proto, .md, .txt, .yml, .yaml, .json, .png, .jpg, .jpeg, .svg, .gif, .html, .sh |
| Kotlin | .kt, .kts | .kt, .kts, .java, .xml, .gradle, .properties, .md, .txt, .json, .yaml, .yml, .toml, .png, .jpg, .jpeg, .svg, .gif, .html, .sh |
| JavaScript | .js, .jsx | .js, .jsx, .json, .md, .txt, .yml, .yaml, .vue, .png, .jpg, .jpeg, .svg, .gif, .html, .css, .scss, .sass, .less, .sh |
| C++ | .cpp, .cc, .cxx, .c++, .hpp, .hh, .hxx | .cpp, .cc, .cxx, .c++, .hpp, .h, .hh, .hxx, .c, .cmake, .txt, .md, .json, .yml, .yaml, .mk, .png, .jpg, .jpeg, .svg, .gif, .html, .sh |
| C | .c, .h | .c, .h, .cmake, .txt, .mk, .makefile, .md, .json, .yml, .yaml, .png, .jpg, .jpeg, .svg, .gif, .html, .sh |
| Rust | .rs | .rs, .toml, .lock, .md, .txt, .png, .jpg, .jpeg, .svg, .gif, .html, .json, .sh |
| Ruby | .rb | .rb, .erb, .rake, .gemspec, .yml, .yaml, .md, .txt, .png, .jpg, .jpeg, .svg, .gif, .html, .json, .sh |
| PHP | .php | .php, .xml, .yml, .yaml, .ini, .md, .txt, .png, .jpg, .jpeg, .svg, .gif, .json, .html, .sh |
| C# | .cs | .cs, .csproj, .sln, .json, .xml, .config, .md, .txt, .png, .jpg, .jpeg, .svg, .gif, .html, .sh |

# A. Data Processing Details

In this appendix, we provide the comprehensive implementation details of the data cleaning, filtering, reconstruction, and verification pipeline described in Section 2. The pipeline is implemented to ensure that only high-quality, reproducible, and semantic code changes are included in the Clean-PR dataset. Figure 4 illustrates how Clean-PR bridges the "noise-validity gap" by strictly filtering failure modes and enhancing valid signals.

## A.1. Language Detection and Extension Rules

To ensure the quality of training data, we enforce a strict two-stage filtering pipeline based on file extensions. First, we determine the primary programming language of each Pull Request (PR) by counting the modified files that match the **Core** extensions (e.g., `.py` for Python, `.java` for Java). The language with the highest frequency of core files is assigned to the PR. If a PR modifies no core files (e.g., only documentation changes), it is immediately discarded.

Second, to eliminate noise from binary files or unrelated assets, we apply a rigorous purity check using the **Allowed** set. Once the language is determined, we verify that *every* file modified in the PR possesses an extension listed in the *Allowed* set for that language (enumerated in Table 15). Crucially, this is a PR-level constraint: if a PR contains any file not listed in the Allowed set, the entire PR is dropped. This ensures that our model is not exposed to PRs containing ambiguous or non-textual artefacts. Finally, for valid PRs, we retain only the files matching the *Core* extensions for training to focus on code logic changes.

## A.2. PR Validity and Noise Filtering

To distil high-quality editing signals from noisy GitHub data, we apply a multi-stage filtering pipeline. A PR is discarded if it triggers any of the following exclusion criteria:

**1. Automation and Bot Filtering.** We exclude PRs created by automated tools or where the only activity comes from bots. A user is classified as a bot if their username matches any of the following regular expressions:

- **Suffix Patterns:** `bot$`, `_bot$`, `-bot$`

- **Prefix Patterns:** `^bot`

- **Specific Services:** `dependabot`, `renovate`, `github-actions`, `travis-ci`, `circleci`, `coveralls`, `auto`, `automated`

**2. Metadata and Quality Constraints.** We filter PRs based on status and textual content to ensure semantic relevance:

- **Status:** The PR must be marked as `MERGED` or `APPROVED`.

- **Title Blocklist:** We remove maintenance PRs containing keywords: `bump`, `dependencies`, `dependency`, `depend`, `release`.

- **Description Blocklist:** We remove descriptions containing `qwiet` (indicating automated security scans).

- **Length Heuristics:** To ensure sufficient context, Titles must be $\geq 10$ characters, and Descriptions must be $\geq 20$ characters.

**3. Structural Integrity Checks.** We strictly enforce that the PR represents a clean, in-place modification of existing code. We exclude PRs that:

- **Missing Base Code or Diff:** PRs with empty base code files or missing diffs are filtered out.

- **Mismatched Files:** Have a discrepancy between the set of files in the base state and the files in the diff (i.e., bijective mapping required).

## A.3. Bug Identification and Issue Linking

We augment PRs with context from linked Issues. We extract issue numbers from titles and descriptions using the following prioritised regex patterns:

1. **Hash References:** `#(\d+)`
2. **Explicit Keywords:**
   - `issue[:\s#-]*(\d+)`
   - `bug[:\s#-]*(\d+)`
   - `fix(es)?[:\s#-]*(\d+)`
   - `resolve(s|d)?[:\s#-]*(\d+)`
   - `close(s|d)?[:\s#-]*(\d+)`
   - `gh-(\d+)`

## A.4. Verified Search/Replace Conversion Pipeline

We convert raw unified diffs into deterministically verifiable **Search/Replace** blocks through a three-stage pipeline, as detailed in Algorithm 1.

**1. Ground-Truth Reconstruction (Fake Git Apply).** We reconstruct the "After" state using a git sandbox to handle context fuzzy-matching.

1. Initialise a temporary git repository with the base code.
2. Apply the diff hunk using `git apply` with fallback strategies:
   - `--verbose`

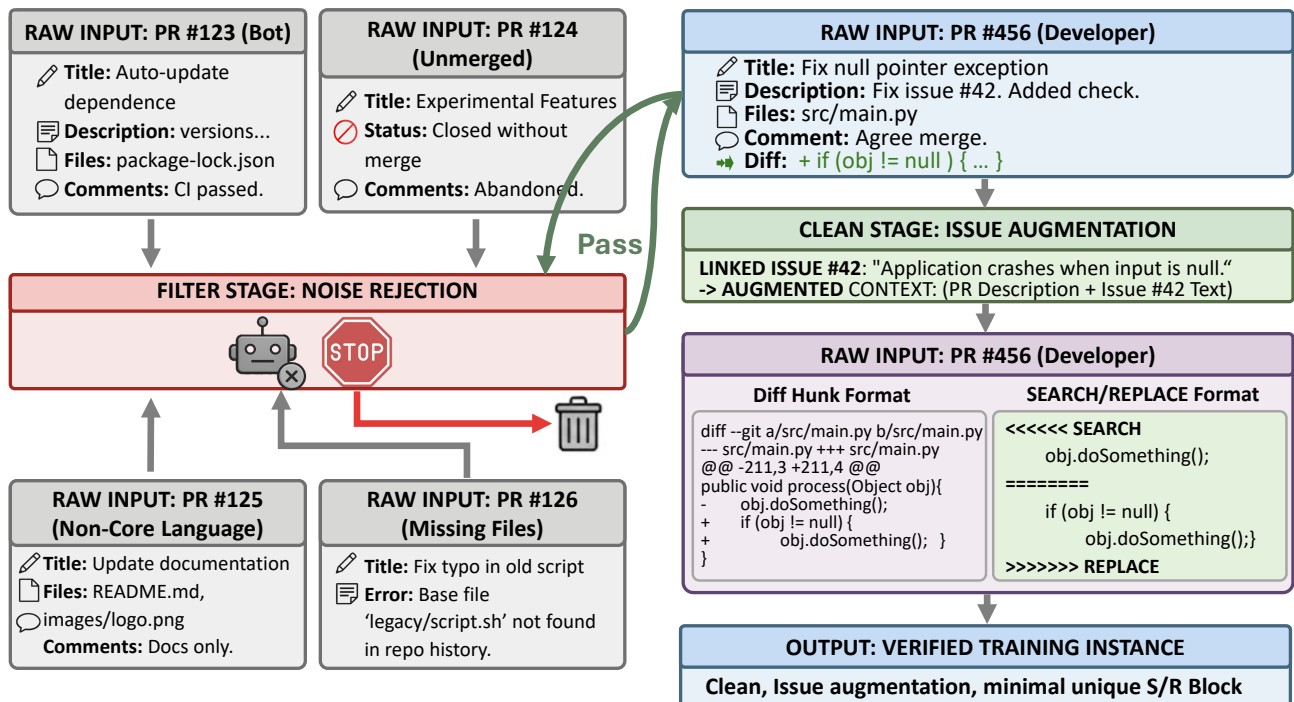

**Figure 4. The Life of a Data Point: From Raw Noise to Verified Signal. Track A (Left)** illustrates the aggressive pruning of noise, rejecting inputs due to bot activity, unmerged status, non-core language files, or missing history. **Track B (Right)** depicts the transformation of a valid PR: it is **augmented** with the linked Issue context to recover user intent and **converted** into a deterministic *Search/Replace* block for verifiable training.

- `--ignore-whitespace`
- `--ignore-space-change`
- `--whitespace=fix`

3. If the application fails, the PR is discarded. If successful, the result defines the `expected_content`.

**2. Minimal Unique Context Search.** We identify edit spans by computing the difference between the base and the reconstructed target files. To generate `SEARCH` blocks that are both concise and unambiguous, we employ an iterative expansion strategy:

- **Edit Merging:** Adjacent edits (separated by $\leq 1$ line) are coalesced into a single block to maintain semantic continuity.

- **Context Expansion:** For each edit, we initialise a context window of size zero. We iteratively expand this window symmetrically: adding lines above and below the edit until the resulting `SEARCH` block occurs **exactly once** within the full file. This guarantees that the model learns the *minimum* context necessary for unique localisation.

**3. Round-Trip Verification.** We validate the generated blocks by performing a strict "round-trip" application using

simple string replacement, independent of git. A training instance is retained only if it passes three integrity checks:

1. **Uniqueness:** Each generated `SEARCH` block must be found exactly once in the base file.
2. **Non-Overlapping:** Multiple edit blocks within the same file must not have overlapping search regions.
3. **Exact Reconstruction:** Applying the Search/Replace blocks to the base file via string replacement must yield a file that is **bit-wise identical** to the ground-truth `target_content` derived in Step 1.

### A.5. Context Windowing Strategy

For files exceeding the token limit (e.g., 100k tokens), we employ a focus-and-expand strategy:

1. **Identify Ranges:** Extract line ranges $[start, end]$ covered by verified Search/Replace blocks.
2. **Expand:** Extend each range by $N = 20$ lines to capture local definitions.
3. **Merge & Reconstruct:** Merge overlapping ranges and concatenate them, inserting markers for omitted sections.

This ensures the model sees the necessary context for the

edit without processing the entire file.

### A.6. Rigorous Decontamination Protocol

To ensure the integrity of our evaluation on SWE-bench and address potential leakage via code propagation (e.g., forks, vendored dependencies), we enforce a multi-layered decontamination pipeline.

**1. Repository-Level Exclusion.** As a primary defence, we strictly blocklist all repositories present in the SWE-bench Lite and Verified metadata. Any Pull Request originating from or targeting these repositories is structurally discarded.

**2. Content-Based Decontamination (Addressing Code Movement).** Relying solely on repository names is insufficient due to the prevalence of code cloning and vendored directories. To mitigate this, we implement content-aware filtering:

- **Exact File Matching:** We compute SHA-256 hashes for all source files in the training corpus. If any file strictly matches a file version found in the evaluation set (spanning the entire test timeline), the instance is flagged. This effectively catches copied or moved code regardless of the repository it resides in.

- **N-gram Overlap:** For partial matches, we index all *Gold Patches* and *Issue Descriptions* from the test set. We exclude training instances that share a **15-gram** code subsequence with gold patches or exceed a **0.5 Jaccard similarity** with issue descriptions, following established protocols (Kocetkov et al., 2023).

### A.7. Semantic Leakage Analysis

Beyond lexical decontamination, we further test whether residual train-test similarity explains Clean-PR's SWE-bench Verified performance. Following the contamination analysis framework of Riddell et al. (2024), we evaluate the hypothesis that leaked or near-leaked examples should make more similar test instances easier to solve.

For each Clean-PR training sample, we concatenate the issue description and code patch; for each SWE-bench Verified instance, we concatenate the problem statement and gold patch. We embed the Python subset of Clean-PR-train (approximately 390K samples) and all 500 SWE-bench Verified instances using BGE-Code-v1 (Li et al., 2025), build a FAISS IndexFlatIP index (Johnson et al., 2021), and assign each test instance its maximum nearest-neighbour similarity score. We then split the evaluation set into five equal-sized similarity quintiles.

The pattern does not support the leakage hypothesis: the most similar quintile has the lowest resolve rate (25.0%),

*Table 16.* SWE-bench Verified resolution rate by maximum train-test similarity quintile.

| Similarity Quintile | Sim Range | # Inst. | Resolve Rate |
|---|---|---|---|
| Q5 (most similar) | [0.76, 0.94) | 100 | 25.0 |
| Q4 | [0.72, 0.76) | 100 | 33.0 |
| Q3 | [0.67, 0.72) | 100 | 34.0 |
| Q2 | [0.61, 0.67) | 100 | 30.0 |
| Q1 (least similar) | [0.35, 0.61) | 100 | 31.0 |
| **Overall** | – | **500** | **30.6** |

while the least similar quintile reaches 31.0%. The Pearson correlation between maximum similarity and binary resolve outcome is $r = -0.061$ with $p = 0.184$, which is not statistically significant. We also manually inspected the top-10 most similar train-test pairs and found common coding idioms or library-level patterns rather than task-specific leakage. Combined with the repository exclusion, SHA-256 matching, 15-gram code filtering, and issue Jaccard filtering above, this suggests that the observed gains are better explained by generalisation than by contamination.

*Table 17.* Language distribution for **Clean-PR-full** (Pre-sampling).

| Language | Count | Ratio (%) | Tokens (B) |
|---|---|---|---|
| Python | 543,419 | 17.81 | 7.77 |
| C++ | 235,246 | 7.71 | 7.45 |
| Go | 409,859 | 13.43 | 6.80 |
| Java | 454,981 | 14.91 | 6.17 |
| JavaScript | 371,640 | 12.18 | 4.55 |
| Rust | 239,346 | 7.85 | 4.12 |
| TypeScript | 278,881 | 9.14 | 3.07 |
| C | 81,789 | 2.68 | 2.29 |
| Kotlin | 132,316 | 4.34 | 1.15 |
| C# | 88,990 | 2.92 | 1.11 |
| PHP | 64,526 | 2.12 | 0.96 |
| Ruby | 149,946 | 4.91 | 0.94 |
| **Total** | **3,050,939** | **100.00** | **46.38** |

### A.8. Language Distribution

We support 12 major programming languages. Table 17 and Table 18 detail the distribution of instances and tokens for the Full and Train sets, respectively. The filtering process preserves the relative diversity of languages, with Python, Go, and C++ remaining the dominant contributors.

### A.9. Data Formatting

**Input Sequence Template.** Table 19 illustrates the exact string formatting template used to construct the Mid-training sequences. We linearise the repository context, issue de-

*Table 18.* Language distribution for **Clean-PR-train** (Post-sampling). This dataset is used for mid-training.

| Language | Count | Ratio (%) | Tokens (B) |
|---|---|---|---|
| Python | 389,881 | 19.34 | 3.83 |
| Go | 268,302 | 13.31 | 2.33 |
| C++ | 154,346 | 7.66 | 2.33 |
| JavaScript | 269,176 | 13.35 | 2.04 |
| Java | 248,251 | 12.32 | 1.91 |
| Rust | 150,024 | 7.44 | 1.52 |
| TypeScript | 188,690 | 9.36 | 1.22 |
| C | 56,812 | 2.82 | 0.76 |
| Ruby | 109,640 | 5.44 | 0.54 |
| C# | 58,045 | 2.88 | 0.45 |
| Kotlin | 78,238 | 3.88 | 0.40 |
| PHP | 44,303 | 2.20 | 0.35 |
| **Total** | **2,015,708** | **100.00** | **17.67** |

scription, and code base into a unified text stream, followed by the target Search/Replace edits.

*Table 19.* The linearised input template used for Mid-training.

```
Clean-PR Format

Repository Name: {repo_name}
Pull Request title: {pr_title}
Description:
{pr_description}

Pull Request codes:
{base_code_content}

SEARCH/REPLACE edits:
{search_replace_format}

Comments:
{valid_comments}
```

### A.10. Data Release Specifications

To ensure full reproducibility and facilitate downstream analysis, we will release the Clean-PR dataset with comprehensive metadata. Table 20 details the definition of each field, including repository metadata, statistical metrics (e.g., token counts), and processing flags (e.g., windowing usage).

## B. StarCoder2-style Data Construction

To ensure a fair comparison, we constructed a strong baseline dataset rigorously following the data processing pipeline of StarCoder2 (Lozhkov et al., 2024). Starting from our raw collection of **16.4 million** crawled Pull Requests (PRs), we applied a multi-stage filtering, sampling, and formatting protocol.

### B.1. Filtering and Cleaning Pipeline

We implemented a cascade of filters targeting PR metadata, file content, and text quality.

**PR-level Filtering.** We discard PRs that satisfy any of the following criteria:

- **Bot Activity:** PRs opened by bots or containing comments exclusively from bots (identified by username patterns and keywords).

- **Licence & Status:** PRs from repositories with non-permissive licences (e.g., GPL), user opt-outs, or PRs that were not approved or merged.

- **Integrity:** PRs that change the base branch during the process or lack initial diffs, preventing accurate reconstruction of changes.

**File-level Filtering.** For the files involved in each PR, we apply strict quality controls:

- **Size Constraints:** Files exceeding 1MB in size, 100,000 lines, an average line length $> 100$, or a maximum line length $> 1,000$ are removed.

- **Content Quality:** Files with $< 25\%$ alphanumeric characters or $> 25\%$ hexadecimal characters are discarded to remove binary or obfuscated files. Non-English Markdown files are also excluded.

**Text Cleaning.** To ensure high-quality natural language supervision:

- **Length & Keywords:** We remove PRs with titles $< 10$ characters (or containing generic terms like "dependency", "release") and descriptions $< 20$ characters (or containing spam keywords like "Qwiet").

- **Truncation:** Titles are truncated to 500 characters. Descriptions are truncated to 80 lines (preserving the first 60 and last 20 lines) or a maximum of 1,000 characters.

- **Comment Sanitization:** We remove auto-generated email replies. Comments shorter than 20 characters are discarded unless they are code review comments. For review comments, associated diff hunks $> 10,000$ characters are truncated. All usernames are anonymized to identifiers like `username_0`.

**Result:** After this rigorous filtering, the dataset was reduced from 16.4M to **6,037,781** valid PRs (a 36.8% pass rate).

*Table 20.* Detailed schema of the released Clean-PR dataset. The corpus retains granular metadata and statistics to support diverse research directions beyond direct training.

| Category | Field Name | Description |
|---|---|---|
| **Metadata** | `repo_name` | The identifier of the source repository (e.g., `owner/repo`). |
| | `repo_url` | The persistent URL to the GitHub repository for attribution. |
| | `detected_language` | The primary programming language of the modified files (e.g., Python). |
| | `is_use_windows` | Boolean flag indicating if the base code was truncated/windowed. |
| **Content** | `pr_title` | The original title of the Pull Request. |
| | `pr_description` | The detailed issue description or PR body text outlining the intent. |
| | `formatted_text` | The final flattened string sequence constructed using the template in Table 19. |
| **Code Artefacts** | `base_code` | The raw content of the source files *before* the edits are applied. |
| | `diff` | The verified **Search/Replace** block sequence used as the training target. |
| | `valid_comments` | (Optional) Reviewer comments aligned with the code changes, if available. |
| **Statistics** | `token_count` | The total number of tokens in the `formatted_text` (using Qwen2.5 Coder tokenizer). |
| | `changed_files_count` | The number of distinct files modified in this Pull Request. |
| | `diff_lines` | The total number of lines added or removed in the diff hunk. |

**Pull Request Template.** The PR input sequence is constructed as follows:

```
StarCoder2-style PR Format

Pull Request Title: {title}
Created by username_0: {description}
Status: {status}
Repository Name: {repo_name}
Base files:
File: {filepath}
Content: {content}
Diff changes: {diffchange}
Comments:
Comment by {username}: {content}
```

## B.2. Rebalancing and Sampling

To mitigate the over-representation of prolific repositories, we adopt the linear downsampling strategy used in Star-Coder2. Concretely, for a repository containing $n$ valid PRs, we retain PRs with a probability that depends on $n$: when $n = 1$, the retention probability is set to $0.8$; when $1 < n \leq 1000$, the probability decreases linearly from $0.8$ to $0.1$ as $n$ increases; and when $n > 1000$, we set the probability so that, in expectation, exactly 100 PRs are retained from that repository. After applying this sampling procedure, the dataset is further reduced to **2,112,688** high-quality PR instances.

## B.3. Data Formatting

We serialise the PRs and Issues into a unified text format. Unlike our proposed method which uses explicit Search/Replace blocks, the StarCoder2-style baseline uses a descriptive natural language format.

**Issue Template.** We also aggregate linked GitHub Issues using the standard conversation format:

```
StarCoder2-style Issue Format

Title: {title}
Issue: {issue_content}
```

The final StarCoder2-style baseline dataset comprises **17.4 billion** tokens.

*Table 21.* Training configurations for mid-training and SFT.

| Setting | Mid-training | SFT |
|---|---|---|
| Model size | 32B | 32B |
| Precision | BF16 | BF16 |
| DeepSpeed ZeRO-3 | ✓ | ✓ |
| FlashAttention-2 | ✓ | ✓ |
| Liger-Kernel | ✓ | ✓ |
| Optimiser | AdamW | AdamW |
| LR scheduler | Cosine | Cosine |
| Warmup ratio | 0.03 | 0.03 |
| Peak learning rate | $2.0 \times 10^{-5}$ | $5.0 \times 10^{-6}$ |
| Epochs | 2 | 3 |
| Global batch size | 128 | 128 |
| Per-device batch size | 2 | 2 |
| Gradient accumulation | 2 | 2 |
| GPU type | H200 | H200 |
| GPU counts | 32 | 32 |
| Context length | 32,768 | 32,768 |
| Training time (wall-clock) | 259 h | 38 h |

## C. Training and inference configuration

**Inference Framework.** We adopt a **Simplified Agentless** scaffolding for evaluation, which mirrors our training alignment by decomposing the resolution process into three deterministic steps: (1) **File localisation** (Table 22), (2) **Line-level navigation** (Table 23), and (3) **Patch Generation** (Table 24). We utilise default decoding parameters (greedy decoding with temperature 0) to ensure reproducibility. For the experiments in Section 3.3, we set the temperature to 0.8. Crucially, to optimise the context window usage, we enforce strict retrieval constraints: for the downstream **Context Construction** (Step 2) and **Patch Generation** (Step 3) phases, we only retain the **top-3** ranked files identified in the initial localisation step.

**Training configurations.** We train the 32B model with BF16 using DeepSpeed ZeRO-3, FlashAttention-2, and Liger-Kernel optimisations (Table 21). For mid-training, we use AdamW with a cosine learning-rate schedule and a warmup ratio of 0.03, training for 2 epochs with a global batch size of 128 (per-device batch size 2 with 2 gradient-accumulation steps) and a peak learning rate of $2.0 \times 10^{-5}$. The SFT stage inherits the same hardware configuration and context length, but uses a smaller learning rate of $5.0 \times 10^{-6}$ for stable adaptation.

## D. Discussion and Future Work

Our at-most-five-core-files filter retains the majority of verified PRs but under-represents the long tail of broad, cross-cutting repository changes; handling such large edits will require longer-context training data and stronger planning or verification components. Extending Clean-PR to other architectures, tokenizers, and MoE-based frontier backbones is a natural next step but lies beyond the scope of this work.

*Table 22.* Prompt for Step 1: File localisation

---

**Prompt for File localisation**

Please look through the following GitHub problem description and Repository structure and provide a list of files that one would need to edit to fix the problem.

**### GitHub Problem Description ###**
**{problem_statement}**

**###**

**### Repository Structure ###**
**{structure}**

**###**

Please only provide the full path and return at most 5 files.
The returned files should be separated by new lines ordered by most to least important and wrapped with ```
For example:
```
file1.py
file2.py
```

*Table 23.* Prompt for Step 2: Fine-grained Navigation

---

**Prompt for Fine-grained Navigation**

Please review the following GitHub problem description and relevant files, and provide a set of locations that need to be edited to fix the issue.
The locations can be specified as class names, function or method names, or exact line numbers that require modification.

### GitHub Problem Description ###
{problem_statement}

###
{file_contents}

###

Please provide the class name, function or method name, or the exact line numbers that need to be edited.
The possible location outputs should be either "class", "function" or "line".

### Examples:
```
full_path1/file1.py
line: 10
class: MyClass1
line: 51

full_path2/file2.py
function: MyClass2.my_method
line: 12

full_path3/file3.py
function: my_function
line: 24
line: 156
```

Return just the location(s) wrapped with ```.

---

*Table 24.* Prompt for Step 3: Patch Generation

---

**Prompt for Patch Generation**

We are currently solving the following issue within our repository. Here is the issue text:
— BEGIN ISSUE —
**{problem_statement}**
— END ISSUE —

**{repair_relevant_file_instruction}**
— BEGIN FILE —
```
**{content}**
```
— END FILE —

Please first localise the bug based on the issue statement, and then generate *SEARCH/REPLACE* edits to fix the issue.

Every *SEARCH/REPLACE* edit must use this format:
1. The file path
2. The start of search block: <<<<<<< SEARCH
3. A contiguous chunk of lines to search for in the existing source code
4. The dividing line: =======
5. The lines to replace into the source code
6. The end of the replace block: >>>>>>> REPLACE

Here is an example:

```python
### mathweb/flask/app.py
<<<<<<< SEARCH
from flask import Flask
=======
import math
from flask import Flask
>>>>>>> REPLACE
```

Please note that the *SEARCH/REPLACE* edit REQUIRES PROPER INDENTATION. If you would like to add the line ' print(x)', you must fully write that out, with all those spaces before the code!
Wrap the *SEARCH/REPLACE* edit in blocks ```python...```.

---

**Algorithm 1** Specification Compatibility Checking

---

**Input** : Base File Content $C_{base}$, Raw Diff Hunk $D$
**Output** : Set of Verified Blocks $\mathcal{S}$ or Failure $\perp$

// Phase 1:  Ground Truth Reconstruction

1   $C_{target} \leftarrow$ FakeGitApply$(C_{base}, D)$
2   **if** $C_{target}$ *is Invalid* **then**
3     **return** $\perp$

// Phase 2:  Minimal Unique Context Search

4   $\Delta \leftarrow$ ComputeDiffOps$(C_{base}, C_{target})$
5   $\mathcal{S} \leftarrow \emptyset$
6   **for** *edit operation* $\delta \in \Delta$ **do**
7     Let $[s, e]$ be the line range of $\delta$ in $C_{base}$
8     $k \leftarrow 0$ // Init context size
9     **for** $k$ **in** *range(0, MAX_CONTEXT)* **do**
      // Expand window symmetrically
10       $start \leftarrow \max(0, s - \lfloor k/2 \rfloor)$
11       $end \leftarrow \min(\text{Len}(C_{base}), e + \lceil k/2 \rceil)$
12       $S_{search} \leftarrow C_{base}[start : end]$
      // Check uniqueness in full file
13       **if** $C_{base}.Count(S_{search}) == 1$ **then**
14         $S_{replace} \leftarrow$ GetNewContent$(C_{target}, \delta)$
15         $\mathcal{S}$.add$(\{S_{search}, S_{replace}\})$
16         **break**

// Phase 3:  Round-Trip Verification

17   $C_{verify} \leftarrow C_{base}$
18   **for** *each block* $B \in \mathcal{S}$ **do**
    // Deterministic String Replacement
19     $locs \leftarrow$ FindIndices$(B.search, C_{verify})$
20     **if** $Length(locs) \neq 1$ **then**
21       **return** $\perp$ // Safety check failed
22     $C_{verify} \leftarrow$ Replace$(C_{verify}, B.search, B.replace)$

// Bit-wise Equality Check

23   **if** $C_{verify} == C_{target}$ **then**
24     **return** $\mathcal{S}$
25   **else**
26     **return** $\perp$ // Artefacts detected

---

