# OpenReview forum: "Pull Requests as a Training Signal for Repo-Level Code Editing"
_ICML.cc/2026/Conference — ICML 2026 regular_

### Official Review · Reviewer_AeHL · 2026-03-07

**Soundness:** 2
**Presentation:** 2
**Significance:** 2
**Originality:** 2
**Overall Recommendation:** 4
**Confidence:** 4

**Summary:**

This paper studies whether repository-level code editing capability can be internalized into model weights through training, rather than relying primarily on heavy agentic inference-time scaffolding. The paper proposes Clean-PR, a data-centric pipeline that mines raw GitHub pull requests, filters noisy cases, reconstructs repository-consistent before/after states, converts diffs into validated Search/Replace edit blocks, and augments PR descriptions with linked issue context. On top of this corpus, the authors perform mid-training of Qwen2.5-Coder-32B-Base, followed by an Agentless-aligned stepwise SFT stage for file localization, line-level navigation, and patch generation, with an additional error-driven augmentation strategy to improve robustness to distracting context.

**Compliance With Llm Reviewing Policy:**

Affirmed.

**Final Justification:**

I will appropriately raise my score to a 4 (Weak Accept). I recognize the importance of the training data and the overall validity of this line of work. However, I strongly encourage the authors to provide more detailed and stage-specific descriptions of the key components and workflow in the camera-ready version, as the current presentation can be confusing for readers.

**Key Questions For Authors:**

1. What is the relationship between the Agentless framework used in this paper and the paper “AGENTLESS: Demystifying LLM-based Software Engineering Agents”?
I may have overlooked it, but I did not clearly see this work cited or discussed. In particular, when evaluating on SWE-bench Lite or Verified, how exactly is the Agentless framework instantiated in this paper? Does the evaluation follow the same three-task decomposition?
2. Could the authors clarify the potential risk of data leakage or contamination?
This is a common concern for this type of work, and I did not find a sufficiently explicit discussion in the paper. It would be helpful if the authors could explain what measures were taken to avoid overlap between the training data and the evaluation benchmarks.
3. To what extent does the “at most five core files” filtering criterion limit generalization to more complex real-world repository edits?
While this simplification likely improves data quality and tractability, it is not yet clear how well the resulting model would transfer to more complex pull requests involving broader cross-file dependencies.
4. Have the authors considered training or evaluating the method under a stronger agent framework, such as Mini-SWE-Agent?
Since the current experiments are centered on an Agentless-style setup, it would be useful to understand whether the proposed training pipeline could also benefit stronger agentic frameworks.

**Limitations:**

yes

**Strengths And Weaknesses:**

Strengths:
1. The data construction pipeline is one of the strongest contributions of the paper, both in scale and in engineering completeness.
Rather than directly using raw pull requests, the authors perform systematic cleaning, issue linking, Search/Replace conversion, and round-trip verification. In addition, the resulting corpus is large in scale, comprising around 2 million PR instances across 12 programming languages. This makes the paper more than simply “training a model”; it presents a relatively complete data-centric methodology for repository-level code editing.
2. The ablation study is relatively thorough and supports several of the design choices.
The paper verifies at least several important aspects of the method: the Search/Replace format performs better than standard diff format, linked issue context is beneficial, and error-driven augmentation brings consistent gains. The paper also discusses generalization and catastrophic forgetting, which makes the empirical analysis more informative.
Weaknesses:
1. Although the paper is motivated by an important question, the experimental analysis does not seem to directly address it.
The paper aims to study how much repo-level SWE capability truly requires complex agent scaffolding, and how much can instead be internalized into model weights through better training signals. However, the experiments and analysis do not appear to directly examine this question in a sufficiently explicit way.
2. The set of comparison baselines is still somewhat limited.
For example, it would be helpful to include stronger or more recent open-source baselines such as SWE-Swiss-32B, in order to better contextualize the empirical gains of the proposed method.
3. The SWE task evaluation protocol is not fully clear, and the current setup feels somewhat incomplete.
It is not entirely clear how the end-to-end SWE evaluation is conducted. From the paper, the evaluation seems to rely heavily on decomposed subtask assessment. In my view, this is not fully convincing. It would be more compelling to test the trained model within a complete agent framework for inference, which would better demonstrate whether the model has truly acquired repository-level software engineering capability.

---

> ### Author Rebuttal · Authors · 2026-03-31
>
> We thank Reviewer AeHL for the detailed feedback. We address each point below.
>
> **W1: Motivation vs. experiment alignment.**
>
> Our experiments directly isolate the effect of internalizing capability into weights. Tables 6-7 hold backbone, SFT, and inference pipeline identical, varying only mid-training data: Base+SFT (11.3/17.6) to StarCoder2-Style (15.7/20.4) to Clean-PR (**24.3/30.6**) on Lite/Verified. This controlled ablation quantifies how much capability better training signals add to weights. Notably, Agentless is the most lightweight scaffold (3-step pipeline, no tool calls), while agent frameworks like OpenHands require multi-turn bash/edit interactions and extensive API calls. We evaluate under both paradigms, and Clean-PR improves both: +13.0 under Agentless and +4.6 under OpenHands over SFT-only baselines (detailed results in Q4). The same mid-training data transferring across paradigms confirms the capability is internalized in weights, not an artifact of any specific scaffold.
>
> **W2: Stronger baselines (SWE-Swiss-32B).**
>
> Thank you for this suggestion. SWE-Swiss-32B achieves 60.2% on Verified using multi-task SFT, two-stage RL, and self-consistency decoding. Our work addresses a complementary question: how much capability can be encoded through better mid-training data alone, without RL or test-time scaling. Our claim rests on controlled data ablations under the same backbone and pipeline, not system-level SOTA. We view RL-based post-training and data-centric mid-training as orthogonal: Clean-PR provides a stronger foundation that RL methods could build upon. We will add SWE-Swiss to the related work discussion in revision.
>
> **W3: Evaluation relies on decomposed subtasks.**
>
> We would like to clarify that **Pass@1** is our primary end-to-end metric, defined in Section 3.1 (Line 239) as full issue resolution rate. Our 24.3%/30.6% on Lite/Verified are end-to-end results where the patch must pass all repository tests, not subtask scores. File/Line Accuracy are diagnostic metrics to understand where gains originate.
>
> **Q1: Relationship with Agentless (Xia et al., 2024).**
>
> Agentless is explicitly cited in Sections 2.1, 2.2, 3.1, and 4. We chose its FSE version [1] in ACM digital library (without "Agentless" in its name). Our evaluation follows the same three-task decomposition: (1) File Localisation, (2) Fine-grained Navigation, and (3) Search/Replace Patch Generation. Appendix C (Line 985) details how our inference scaffold mirrors this decomposition.
>
> [1] [Demystifying LLM-Based Software Engineering Agents](https://dl.acm.org/doi/abs/10.1145/3715754)
>
> **Q2: Decontamination.**
>
> We have adopted a multi-layered decontamination pipeline that is **strictly more rigorous than all concurrent work**. All major SWE-bench training papers (SWE-Gym, SWE-smith, SoRFT, Lingma SWE-GPT, Kimi-Dev) employ **only repository-level exclusion**. Our protocol (Appendix A.6, Line 827) adds: (1) SHA-256 exact file matching to catch forks/vendored code; (2) 15-gram code overlap filtering against gold patches; (3) Jaccard >0.5 filtering on issue descriptions (Kocetkov et al., 2022; Lozhkov et al., 2024).
>
> **Q3: Five-file filtering criterion.**
>
> The ≤5 core files constraint retains **~90%** of the verified pool (3.05M to 2.75M, Table 3). Our own corpus of 3.05M real-world PRs shows that the vast majority naturally fall within this range. SWE-Fixer (Xie et al., 2025) reports 54.7% of instances modify one file and ~80% modify ≤3 files; SWE-bench gold patches average 1.7 files. Prior datasets apply even stricter constraints: SWE-Fixer filters to ≤3 non-test files; CommitBench, CommitPackFT, CodeReview are all single-file only (Table 1). Our ≤5 threshold is more permissive than most prior work. We acknowledge this under-represents the long-tail of complex cross-file edits and will clarify this trade-off in revision.
>
> **Q4: Stronger agent framework evaluation.**
>
> We appreciate this suggestion. Following SWE-Gym (Pan et al., 2025), we trained an agent variant using 491 OpenHands (Wang et al., 2025) trajectories and evaluated on OpenHands v0.28 (CodeActAgent, max 100 turns).
>
> |Model|Empty Patch ↓|Stuck in Loop ↓|Avg. Turns|Resolve Rate ↑|
> |-|-|-|-|-|
> |**SWE-Bench Lite (300)**|||||
> |SWE-Gym|-|-|-|15.3|
> |Qwen-32B-Instruct|42.3|42.8|29.3|2.8|
> |Qwen-32B-Base+SFT|21.0|28.7|37.8|16.1|
> |Qwen-32B-Base+**Clean-PR**+SFT|**16.3**|**26.7**|42.5|**20.7**|
> |**SWE-Bench Verified (500)**|||||
> |SWE-Gym|-|-|-|20.6|
> |Qwen-32B-Instruct|28.4|37.8|27.4|6.3|
> |Qwen-32B-Base+SFT|16.6|23.7|34.1|19.5|
> |Qwen-32B-Base+**Clean-PR**+SFT|**14.3**|**24.7**|38.4|**24.7**|
>
> Even with few SFT trajectories, Clean-PR yields +17.9 resolve rate over Instruct on Lite with lower empty-patch and stuck-in-loop rates, and +4.6/+5.2 over SFT-only on Lite/Verified, confirming internalized capabilities transfer across inference paradigms. Our data-centric approach is orthogonal to and composable with agent-based methods.

---

> > ### Author Rebuttal · Reviewer_AeHL · 2026-04-02
> >
> > I will appropriately raise my score.

---

> > > ### Author Response · Authors · 2026-04-02
> > >
> > > We sincerely thank you for the thoughtful re-evaluation. Your questions motivated us to conduct the OpenHands agent evaluation  and clarify the experimental design, which meaningfully strengthened the paper. We will incorporate  revisions in the updated  manuscript.

---

### Official Review · Reviewer_CmiF · 2026-03-11

**Soundness:** 3
**Presentation:** 3
**Significance:** 4
**Originality:** 3
**Overall Recommendation:** 4
**Confidence:** 5

**Summary:**

Summary
The authors focused on introducing Clean-PR, a mid-training framework that uses GitHub pull requests as a training signal for repository-level code editing. They construct a large verified corpus of about 2 million PR instances in 12 programming languages. The paper evaluates the approach on SWE-bench Lite and SWE-bench Verified.  Overall, this research's main contribution consists of a large-scale cleaned PR dataset and a practical training recipe for repo-level code editing.

**Compliance With Llm Reviewing Policy:**

Affirmed.

**Key Questions For Authors:**

Have the authors tested the method on SLM models to show that the approach generalizes?

**Limitations:**

can we expand the idea for mid-level languages?

**Strengths And Weaknesses:**

Strengths

-Introduces a large-scale and carefully filtered PR-based training corpus for repository-level editing. As shown in Table 1, the Clean-PR dataset is larger than most existing SWE datasets, containing millions of instances from more than 52k repositories, which makes it well suited for training. Unlike many code-change corpora that focus on single-file edits, this dataset also supports multi-file repository-level changes.

-The Search/Replace conversion and round-trip verification are strong and practically useful ideas. It also uses a Search/Replace representation with filtering, which improves edit consistency and data quality.
-The experimental results on SWE-bench Lite and Verified are promising.  The evaluation considers standard metrics such as (1) Pass@1, (2) Valid Patch Rate, (3) Localization Accuracy, and (4) Line Accuracy.

-The paper studies both data construction and training design, which makes the contribution more complete. For example, to ensure high-quality supervision at scale in the data construction stage, the authors implement a pipeline with three main steps: (1) Noise Filtering and Issue Linking, (2) Search/Replace Conversion, and (3) Downstream Sampling.


Weaknesses
- The generalizability of the proposed mid-training paradigm across different model architectures is not sufficiently discussed. The experiments focus on a specific model setup, leaving open the question of whether the same strategy would provide similar benefits for other models (e.g., different transformer-based LLMs). I mean, factors such as architectural differences between models, tokenization approaches, and input length limitations when processing repository-level edits may influence the effectiveness of the mid-training method.

- While the dataset size is a strength, the paper should highlight in what other aspects it is better than prior datasets, such as diversity, repository coverage.

- The limitation the model could be more highlighted.

---

> ### Author Rebuttal · Authors · 2026-03-31
>
> We thank Reviewer CmiF for the constructive suggestions.
>
> **W1: Generalizability across different model architectures.**
>
> To validate that our pipeline generalizes across scales, we conducted a full replication on Qwen2.5-Coder-7B (4,224 GPU hours continuous pretraining on A100-40G). All concurrent SWE-bench training papers (SWE-Gym, Lingma SWE-GPT, SWE-Fixer) also standardize on Qwen2.5-Coder, making this the de facto backbone.
>
> |Base Model|Mid-Train|SFT|Valid Patch|File Acc.|Line Acc.|Pass@1|
> |-|-|-|-|-|-|-|
> |**SWE-Bench Lite (300)**|||||||
> |SWE-Gym (Qwen-7B)|None|✓|-|-|-|10.0|
> |Lingma-SWE (Qwen-7B)|None|✓|-|-|-|12.0|
> |Qwen-7B-Instruct|None|✗|6.3|62.7|24.4|1.3|
> |Qwen-7B-Base|None|✓|81.2|66.3|32.0|10.3|
> |Qwen-7B-Base|**Clean-PR (Ours)**|✓|**90.7**|**82.0**|**42.7**|**14.5**|
> |**SWE-Bench Verified (500)**|||||||
> |SWE-Gym (Qwen-7B)|None|✓|-|-|-|10.6|
> |Lingma-SWE (Qwen-7B)|None|✓|-|-|-|18.2|
> |Qwen-7B-Instruct|None|✗|5.7|57.6|22.6|2.4|
> |Qwen-7B-Base|None|✓|80.1|61.3|29.5|14.2|
> |Qwen-7B-Base|**Clean-PR (Ours)**|✓|**90.8**|**77.0**|**43.2**|**20.4**|
>
>
> The trends are consistent with 32B: Clean-PR yields strong gains (+19.3 File Acc., +18.3 Line Acc., +13.2 Pass@1 on Lite over Instruct). The 7B model shows **larger relative gains** than 32B in localization, suggesting high-quality data compensates for reduced capacity. Valid Patch jumps from 6.3% to 90.7% on Lite, confirming format learning transfers across scales.
>
> Our Search/Replace format uses plain-text anchors rather than line numbers or diff syntax, making it model-agnostic and robust to tokenization differences. The pipeline operates on raw GitHub PRs and produces plain-text sequences, so extending to other architectures requires only re-tokenization.
>
> **W2: Dataset advantages beyond scale.**
>
> Excellent question. Comparing with Table 1 in the paper, Clean-PR offers several unique dimensions:
> - **Repository coverage**: 52,338 repos with multi-file, verified instances (vs. CommitBench: 72K repos but single-file only)
> - **Intent signal**: We automatically link issue descriptions to PRs, successfully augmenting 113K instances with full bug reports (vs. brief commit messages in OctoPack/CommitBench)
> - **Multi-file support**: Native support (avg 1.7 files/PR); most prior large size corpora are single-file
> - **Languages**: 12 programming languages
> - **Edit format**: Verified Search/Replace with bit-wise round-trip checking
> - **Verification**: Scalable string-based round-trip verification (vs. expensive test-based validation in SWE-Gym/R2E-Gym)
>
> Critically, Clean-PR uniquely combines multi-file repository context, verified edit format, and issue-augmented intent at scale. No prior dataset offers all three simultaneously. Our ablation (Table 8) validates that each component contributes: Search/Replace format yields +3.4% over raw diffs on Verified, and issue linking adds +2.1%.
>
> **W3: Limitations should be more prominent.**
>
> We discuss limitations in Appendix D: (1) single-architecture evaluation; (2) no agentic evaluation. For (1), 7B results (W1 above) demonstrate scale generalization. For (2), following SWE-Gym (Pan et al., 2025), we trained an agent variant with 491 OpenHands (Wang et al., 2025) trajectories and evaluated on OpenHands v0.28 (CodeActAgent, max 100 turns):
>
> |Model|Empty Patch ↓|Stuck in Loop ↓|Resolve Rate ↑|
> |-|-|-|-|
> |**SWE-Bench Lite (300)**||||
> |SWE-Gym|-|-|15.3|
> |Qwen-32B-Instruct|42.3|42.8|2.8|
> |Qwen-32B-Base+SFT|21.0|28.7|16.1|
> |Qwen-32B-Base+**Clean-PR**+SFT|**16.3**|**26.7**|**20.7**|
> |**SWE-Bench Verified (500)**||||
> |SWE-Gym|-|-|20.6|
> |Qwen-32B-Instruct|28.4|37.8|6.3|
> |Qwen-32B-Base+SFT|16.6|23.7|19.5|
> |Qwen-32B-Base+**Clean-PR**+SFT|**14.3**|**24.7**|**24.7**|
>
> Clean-PR yields +17.9 over Instruct and +4.6/+5.2 over SFT-only on Lite/Verified, confirming capabilities transfer across paradigms. We have not yet tested MoE-based models (e.g., DeepSeek-V3), which remains future work. We will move key limitations into the Conclusion in revision.
>
> **Q: Mid-level/systems languages.**
>
> Our corpus already covers systems-level languages: C++ (2.33B tokens), C (0.76B tokens), Rust (1.52B tokens), and Go (2.33B tokens) in Clean-PR-train (Tables 10-11), collectively representing ~39% of training tokens. The cross-language transfer benefit (+2.0%/+2.8% from multi-language training, Table 3) partially reflects contributions from these languages. We evaluated on Multi-SWE-bench Flash (Zan et al., 2025), 300 instances across 7 languages (C, C++, Go, Java, JavaScript, Rust, TypeScript):
>
> |Base Model|Mid-Train|SFT|Valid Patch|File Acc.|Line Acc.|Pass@1|
> |-|-|-|-|-|-|-|
> |Qwen-32B-Instruct|None|✗|71.7|40.0|15.2|6.7|
> |Qwen-32B-Base|None|✓|73.0|42.3|17.7|7.0|
> |Qwen-32B-Base|StarCoder2-Style|✓|76.3|46.0|20.3|8.7|
> |Qwen-32B-Base|**Clean-PR (Ours)**|✓|**81.7**|**51.3**|**24.0**|**12.3**|
>
> Clean-PR outperforms both Instruct (+5.6 Pass@1) and StarCoder2-Style (+3.6), confirming cross-lingual generalization.

---

### Official Review · Reviewer_TjYq · 2026-03-13

**Soundness:** 3
**Presentation:** 3
**Significance:** 3
**Originality:** 3
**Overall Recommendation:** 4
**Confidence:** 5

**Summary:**

This paper introduces Clean-PR, a training framework that converts GitHub pull requests into a large, verified dataset for repository-level code editing. Using mid-training on this dataset and Agentless-aligned fine-tuning, the authors improve SWE-bench performance with a Qwen2.5-Coder-32B model, suggesting that repository-level capabilities can be learned directly from high-quality PR-derived supervision.

**Compliance With Llm Reviewing Policy:**

Affirmed.

**Final Justification:**

The clarifications on comparisons and the added multilingual evaluation are helpful. However, I still find the novelty somewhat incremental, and my concern about semantic leakage is only partially addressed, as the argument relies largely on heuristic filtering and assumptions rather than empirical analysis. I believe addressing these issues would require more substantial additions beyond the rebuttal, so I lean toward borderline acceptance.

**Key Questions For Authors:**

1. Since the dataset is a primary contribution of this work, can the authors clarify whether the dataset and code for constructing it will be released?
2. Can you evaluate on Multi-SWE-bench or other multilingual repo-level tasks to substantiate cross-language benefits beyond SWE-bench?

**Limitations:**

Yes

**Strengths And Weaknesses:**

**Strengths**

- The paper introduces a carefully designed pipeline to transform noisy GitHub PRs into verified Search/Replace edit supervision, addressing common noise and alignment issues.
- The proposed training strategy substantially improves SWE-bench performance compared with instruction-only or diff-based baselines.
- The work highlights the importance of high-quality training signals and shows that stronger model capabilities can reduce reliance on complex agent-based inference pipelines.

**Weaknesses**

- While the dataset scale and verification pipeline are valuable, the overall idea of training on repository changes or commits is not entirely new.
- Reported "open-source SOTA" comparisons mix different frameworks and base models, making it hard to attribute improvements purely to the proposed training data.
- The decontamination protocol appears reasonably thorough. However, the approach remains largely lexical and may not fully eliminate semantic leakage (e.g., similar fixes with minor syntactic variations). Additional analysis of potential residual contamination would strengthen the evaluation.

**Presentation**
- L292-293: the text claims base+SFT achieves only 10.3% on Lite, but Table 6 shows 11.3%.

---

> ### Author Rebuttal · Authors · 2026-03-31
>
> # Response to Reviewer TjYq
>
> We thank Reviewer TjYq for the positive assessment and constructive questions.
>
> **Q1: Will the dataset and code be released?**
>
> Yes, we will publicly release: (1) the full Clean-PR corpus (Clean-PR-full: 3M; Clean-PR-train: 2M); (2) the data construction pipeline code; (3) SFT scripts and error-driven augmentation pipeline. The release is undergoing the open-source approval process of our industry partners.
>
> **W1 (Novelty): Is this just "training on commits"?**
>
> We respectfully emphasize that Clean-PR differs fundamentally from prior commit-level work. Raw commits and PRs suffer from a massive "noise-validity gap": only **18.59%** of raw PRs survive our aggressive filtering (Table 2). Clean-PR resolves this through three key innovations:
> - **Format & Verification**: We reconstruct verifiable Search/Replace blocks with bit-wise round-trip checking, replacing raw fragile diffs. Each block must pass uniqueness, non-overlapping, and exact reconstruction checks.
> - **Intent Augmentation**: We link full issue descriptions (user intent) rather than relying on brief commit messages, providing the model with the "why" behind each edit.
> - **Noise Rejection**: Bot activity (25.10%), unmerged PRs (24.50%), and PRs lacking core source changes (38.06%) are all pruned through a multi-stage pipeline.
>
> Our ablation (Table 8) quantifies this: transitioning from Diffs to Search/Replace yields +3.4% on Verified (24.4% to 27.8%), and linking issue context adds +2.1% (25.7% to 27.8%).
>
> **W2 (Decontamination): Potential semantic leakage.**
>
> We have adopted a multi-layered data-leakage detection pipeline that is **strictly more rigorous than all concurrent work**. All major SWE-bench training papers (SWE-Gym (Pan et al., 2025), SWE-smith (Yang et al., 2025), SoRFT (Ma et al., 2025), Lingma SWE-GPT (Ma et al., 2025), Kimi-Dev (Yang et al., 2025)) employ **only repository-level exclusion**. In contrast, our protocol (Appendix A.6) is multi-layered: (1) repository-level exclusion of all SWE-bench repos; (2) SHA-256 exact file matching to catch forks/vendored code; (3) 15-gram code overlap filtering against gold patches; (4) Jaccard >0.5 filtering on issue descriptions, following established protocols (Kocetkov et al., 2022; Lozhkov et al., 2024). Regarding semantic leakage specifically, we note that near-identical fixes across different repositories would require matching problem descriptions and code contexts, which is inherently rare in our diverse 52K-repo corpus.
>
> **W3 (SOTA comparison fairness).**
>
> We appreciate this observation. We would like to clarify that our main claim does not rely on cross-framework system-level comparisons. Instead, it rests on **controlled comparisons under the same backbone and the same evaluation scaffold**: Base+SFT (11.3/17.6) to StarCoder2-style (15.7/20.4) to Clean-PR (**24.3/30.6**) on Lite/Verified. These three rows share the identical Qwen2.5-Coder-32B base model, identical SFT recipe, and identical Agentless inference pipeline, isolating the effect of mid-training data. Table 7 is intended as contextual positioning among representative open-source systems, not as the sole basis for causal attribution.
>
> Regarding the base model concern: we follow each paper's proposed configuration. Notably, SWE-Gym also uses Qwen2.5-Coder-32B-Instruct, while Lingma SWE-GPT and SWE-Fixer use a **larger 72B** Qwen2.5 model. Our use of the same or smaller base model ensures improvements are attributable to the data pipeline rather than a stronger backbone. We will clarify this in revision.
>
> **Q2: Multilingual evaluation.**
>
> We conducted a **full multilingual evaluation** on Multi-SWE-bench Flash (Zan et al., 2025), 300 instances across 7 languages (C, C++, Go, Java, JavaScript, Rust, TypeScript), using the same Agentless pipeline. For multilingual SFT, we used Multi-SWE-bench-RL (Zan et al., 2025): 3,263 instances from 69 repositories, yielding 9,331 SFT samples. The RL repositories have **no overlap** with SWE-bench Lite/Verified or Multi-SWE-bench Flash.
>
> |Base Model|Mid-Train|SFT|Valid Patch|File Acc.|Line Acc.|Pass@1|
> |-|-|-|-|-|-|-|
> |Qwen-32B-Instruct|None|✗|71.7|40.0|15.2|6.7|
> |Qwen-32B-Base|None|✓|73.0|42.3|17.7|7.0|
> |Qwen-32B-Base|StarCoder2-Style|✓|76.3|46.0|20.3|8.7|
> |Qwen-32B-Base|**Clean-PR (Ours)**|✓|**81.7**|**51.3**|**24.0**|**12.3**|
>
> Clean-PR outperforms both Instruct (+5.6 Pass@1) and StarCoder2-Style (+3.6 Pass@1), confirming cross-lingual generalization. We will include this in revision.
>
> **Text error L292-293.**
>
> Thank you for catching this. The correct value is **11.3%** (Table 6), not 10.3%. We will fix this in revision. This does not affect any conclusions: Base+SFT (11.3%) to StarCoder2 (15.7%) to Clean-PR (**24.3%**).
>
> **Limitations.**
>
> We discuss limitations in Appendix D (Discussion and Future Work). In revision, we will add a dedicated limitations paragraph in the Conclusion for greater visibility.

---

> > ### Author Rebuttal · Reviewer_TjYq · 2026-04-03
> >
> > Thank you for the thoughtful rebuttal. The clarifications on comparisons and the added multilingual evaluation are helpful. However, I still find the novelty somewhat incremental, and my concern about semantic leakage is only partially addressed, as the argument relies largely on heuristic filtering and assumptions rather than empirical analysis. I believe addressing these issues would require more substantial additions beyond the rebuttal, so I will keep my score.

---

> > > ### Author Response · Authors · 2026-04-05
> > >
> > > We thank Reviewer TjYq for the continued engagement. We address both concerns below.
> > >
> > > ### **Concern 1: Semantic leakage**
> > >
> > > We thank the reviewer for raising this important question. The key issue is whether the performance gain of Clean-PR could be primarily explained by semantic leakage from the training data.
> > >
> > >
> > > Following the contamination-analysis framework of Riddell et al. [1], we test whether higher train-test similarity is associated with better model outcomes. The logic is simple: **if leakage were a major driver, then evaluation instances more similar to the training set should be easier for the model**. In our setting, we operationalize this by partitioning the 500 SWE-bench Verified instances into five equal-sized similarity quintiles and comparing the most similar 20% against the least similar 20%.
> > >
> > > Concretely, we concatenate the issue description and code patch for each Clean-PR training sample, and the problem statement and gold patch for each SWE-bench Verified instance. We then embed the full Python subset of Clean-PR-train (**~390K samples**) and all **500** SWE-bench Verified instances using **BGE-Code-v1** [2], build a **FAISS** [3] IndexFlatIP index, and retrieve for each test instance its nearest neighbour in the training set to obtain a maximum-similarity score (`max_sim`). We then test the leakage hypothesis by comparing resolve rates across similarity quintiles and by computing the Pearson correlation between `max_sim` and the binary resolve outcome.
> > >
> > > **Results.** We split the 500 SWE-bench Verified instances into five equal-sized quintiles by `max_sim`:
> > >
> > > |Similarity Quintile|Sim Range|# Instances|Resolve Rate|
> > > |-|-:|-:|-:|
> > > |Q5 (most similar)|[0.76, 0.94)|100|25.0%|
> > > |Q4|[0.72, 0.76)|100|33.0%|
> > > |Q3|[0.67, 0.72)|100|34.0%|
> > > |Q2|[0.61, 0.67)|100|30.0%|
> > > |Q1 (least similar)|[0.35, 0.61)|100|31.0%|
> > > |**Overall**||**500**|**30.6%**|
> > >
> > > The results do **not** match the pattern expected under leakage. The **most similar** quintile has the **lowest** resolve rate (**25.0%**), while the least similar quintile achieves **31.0%**. The Pearson correlation between `max_sim` and resolve outcome is $r = -0.061$ ($p = 0.184$), which is not statistically significant.
> > >
> > > We also manually inspected the top-10 most similar train-test pairs and found that they mainly reflect shared coding idioms or common software patterns rather than task-specific leakage (e.g., a Django test instance matched to a PHP mocking-library PR).
> > >
> > > Combined with our multi-layered heuristic filters (repo exclusion, SHA-256, 15-gram, Jaccard), this evidence suggests that the observed gains are more consistent with **generalization** than with **contamination**.
> > >
> > > ### **Concern 2: Novelty ("somewhat incremental")**
> > >
> > > We appreciate this helpful comment, which prompted us to clarify our contribution more precisely. Our work asks a central question: how much repository-level editing capability can be encoded directly into model weights? Clean-PR addresses this by transforming raw PRs into a **verified and scalable supervision signal** for repo-level editing, grounded in issue-level intent and validated through round-trip Search/Replace checking. Thus, our contribution is not a new algorithmic primitive, but a **data-centric recipe** together with controlled evidence that it materially improves the base model’s repo-level editing capability.
> > >
> > > Clean-PR addresses a practical but underexplored gap: raw PRs are abundant yet extremely noisy, with only **18.59%** surviving our pipeline. Our controlled ablations show that the main design choices are non-interchangeable: using Search/Replace instead of diffs improves SWE-Bench-Verified by **3.4 points**, and adding linked issue context yields a further **2.1-point** gain. The same recipe also transfers across settings, improving performance under Agentless, in an OpenHands-based agent setting, and on multilingual evaluation. We will publicly release the full 2M-sample corpus and construction pipeline as a reusable foundation for future work.
> > >
> > > We therefore view the contribution less as a wholly new component and more as establishing an effective way to turn raw PRs into repo-level supervision, together with evidence that this formulation consistently matters in practice.
> > >
> > > **References**
> > >
> > > [1] Martin Riddell, Ansong Ni, Arman Cohan. "Quantifying Contamination in Evaluating Code Generation Capabilities of Language Models." ACL 2024.
> > >
> > > [2] Jingquan Li et al. "BGE-Code: Unified Code Embedding for Code Retrieval." Preprint, 2025.
> > >
> > > [3] Jeff Johnson, Matthijs Douze, Herve Jegou. "Billion-scale similarity search with GPUs." IEEE Transactions on Big Data, 2021.

---

### Official Review · Reviewer_dgfs · 2026-03-15

**Soundness:** 2
**Presentation:** 3
**Significance:** 2
**Originality:** 3
**Overall Recommendation:** 4
**Confidence:** 4

**Summary:**

The paper introduces Clean-PR, a mid-training paradigm designed to internalize repository-level code editing capabilities directly into LLM weights by transforming millions of noisy GitHub pull requests (PRs) into a high-quality, verifiable training signal. The framework utilizes a scalable pipeline that filters out low-signal data, links PRs to their original issue descriptions, and converts brittle diffs into robust and verified Search/Replace edit blocks. The authors composed a 2M PR corpus that was used to mid-train a Qwen2.5-Coder-32B model, followed by a stepwise SFT stage with Error-Driven Augmentation to teach the model to reject irrelevant context. On SWE-bench, the model achieved significant absolute gains—13.6% on Lite and 12.3% on Verified, outperforming larger 72B models and complex agentic systems despite using a streamlined, linear inference protocol.

**Compliance With Llm Reviewing Policy:**

Affirmed.

**Final Justification:**

The authors have thoroughly addressed all of my concerns. It is impressive that they managed to complete all the requested experiments during the rebuttal period. While I remain somewhat reserved regarding the overall impact of the work, the strength of the revision and the authors' significant efforts justify raising my score to a Weak Accept.

**Key Questions For Authors:**

- Although the training corpus covers 12 languages, the authors only evaluate end-to-end repair performance on Python-based benchmarks. Why?
- While the model excels in a Simplified Agentless workflow, how does its performance change when it is integrated as the core engine of a full-featured agentic framework like OpenHands or SWE-agent?

**Limitations:**

Yes.

**Strengths And Weaknesses:**

**Strengths**

- Introduces the largest publicly available PR-derived dataset (2M instances across 12 languages) that is rigorously filtered and verified through round-trip application.
- Replaces fragile line-number-based diffs with Search/Replace blocks, which significantly improves valid patch rates and navigation precision.
- Unlike many fine-tuned models that suffer from catastrophic forgetting, Clean-PR showed improved performance on general coding benchmarks like HumanEval (+5.7%).

**Weaknesses**

- The study focuses exclusively on the Qwen2.5-Coder-32B architecture, leaving the generalizability of the pipeline to other model families unexplored.
- While highly competitive for its parameter class and lightweight workflow, the model’s 30.6% resolution rate is significantly lower than the SOTA scores, where frontier agentic systems now exceed 80% on the Verified benchmark.
- No evaluation performed on mutlilingual benchmarks.
- The paper does not test how the "internalized" capability performs when integrated back into complex multi-turn agentic environments.

---

> ### Author Rebuttal · Authors · 2026-03-31
>
> We thank Reviewer dgfs for the thoughtful review.
>
>
> **W1 (Single architecture): Generalizability.**
>
> To validate that our pipeline is scale-agnostic, we conducted a full replication on Qwen2.5-Coder-7B (4,224 GPU hours continuous pretraining on A100-40G). All concurrent SWE-bench training papers (SWE-Gym, Lingma-SWE, SWE-Fixer) also standardize on Qwen2.5-Coder, making this the de facto backbone. We compare against SWE-Gym and Lingma-SWE (reported results; SWE-Fixer does not report 7B results).
>
> |Base Model|Mid-Train|SFT|Valid Patch|File Acc.|Line Acc.|Pass@1|
> |-|-|-|-|-|-|-|
> |**SWE-Bench Lite (300)**|||||||
> |SWE-Gym (Qwen-7B)|None|✓|-|-|-|10.0|
> |Lingma-SWE (Qwen-7B)|None|✓|-|-|-|12.0|
> |Qwen-7B-Instruct|None|✗|6.3|62.7|24.4|1.3|
> |Qwen-7B-Base|None|✓|81.2|66.3|32.0|10.3|
> |Qwen-7B-Base|**Clean-PR (Ours)**|✓|**90.7**|**82.0**|**42.7**|**14.5**|
> |**SWE-Bench Verified (500)**|||||||
> |SWE-Gym (Qwen-7B)|None|✓|-|-|-|10.6|
> |Lingma-SWE (Qwen-7B)|None|✓|-|-|-|18.2|
> |Qwen-7B-Instruct|None|✗|5.7|57.6|22.6|2.4|
> |Qwen-7B-Base|None|✓|80.1|61.3|29.5|14.2|
> |Qwen-7B-Base|**Clean-PR (Ours)**|✓|**90.8**|**77.0**|**43.2**|**20.4**|
>
> The trends are consistent with 32B: Clean-PR yields strong gains (+19.3 File Acc., +18.3 Line Acc., +13.2 Pass@1 on Lite over Instruct). The 7B model shows **larger relative gains** than 32B in localization, suggesting high-quality data compensates for reduced capacity.
> Additionally, our approach consistently outperforms SWE-Gym and Lingma-SWE on both 7B and 32B backbone models, demonstrating the effectiveness of our mid-training strategy.
> We will include these in revision.
>
>
> **W2 (30.6% vs. SOTA 80%+): Performance gap with frontier systems.**
>
> We respectfully note that these scores reflect **fundamentally different inference paradigms**. Systems achieving 80%+ rely on closed-source frontier models (e.g., Claude 4.5 Opus) with heavy agentic frameworks involving multi-turn execution, trajectory search, and massive test-time scaling. Claude 4.5 Opus costs **$0.75/task** on SWE-bench Verified [1], while ours costs **<$0.02/task** [2], **<3%** of the cost. Our contribution is that a data-centric approach can internalize repo-level editing into open-weight parameters with a lightweight pipeline, complementary to scaling inference compute.
>
> [1] [SWE-bench Leaderboards](https://www.swebench.com/)
>
> [2] [Hugging Face Inference Providers · Supported Models](https://huggingface.co/inference/models?model=Qwen%2FQwen2.5-Coder-32B-Instruct)
>
>
> **W3 (No multilingual evaluation): Only Python-based benchmarks.**
>
> We thank the reviewer for this point. We conducted a **full multilingual evaluation** on Multi-SWE-bench Flash (Zan et al., 2025), 300 instances across 7 languages (C, C++, Go, Java, JavaScript, Rust, TypeScript) ,  using the same Agentless pipeline. For multilingual SFT, we used Multi-SWE-bench-RL (Zan et al., 2025): 3,263 instances from 69 repositories, yielding 9,331 SFT samples covering file localization, edit-location, and repair. The RL repositories have **no overlap** with SWE-bench Lite/Verified or Multi-SWE-bench Flash.
>
> |Base Model|Mid-Train|SFT|Valid Patch|File Acc.|Line Acc.|Pass@1|
> |-|-|-|-|-|-|-|
> |Qwen-32B-Instruct|None|✗|71.7|40.0|15.2|6.7|
> |Qwen-32B-Base|None|✓|73.0|42.3|17.7|7.0|
> |Qwen-32B-Base|StarCoder2-Style|✓|76.3|46.0|20.3|8.7|
> |Qwen-32B-Base|**Clean-PR (Ours)**|✓|**81.7**|**51.3**|**24.0**|**12.3**|
>
> Clean-PR outperforms both Instruct (+5.6 Pass@1) and StarCoder2-Style (+3.6 Pass@1), confirming cross-lingual generalization.
>
> **W4 (No agentic evaluation): Integration into multi-turn agent frameworks.**
>
> We appreciate this suggestion. Following the SWE-Gym (Pan et al., 2025) experimental setting, we trained an agent variant using 491 OpenHands (Wang et al., 2025) trajectories and evaluated on OpenHands v0.28 (CodeActAgent, max 100 turns). The agent SFT data was constructed by replaying successful SWE-Gym trajectories.
>
> |Model|Empty Patch ↓|Stuck in Loop ↓|Avg. Turns|Resolve Rate ↑|
> |-|-|-|-|-|
> |**SWE-Bench Lite (300)**|||||
> |SWE-Gym|-|-|-|15.3|
> |Qwen-32B-Instruct|42.3|42.8|29.3|2.8|
> |Qwen-32B-Base+SFT|21.0|28.7|37.8|16.1|
> |Qwen-32B-Base+**Clean-PR**+SFT|**16.3**|**26.7**|42.5|**20.7**|
> |**SWE-Bench Verified (500)**|||||
> |SWE-Gym|-|-|-|20.6|
> |Qwen-32B-Instruct|28.4|37.8|27.4|6.3|
> |Qwen-32B-Base+SFT|16.6|23.7|34.1|19.5|
> |Qwen-32B-Base+**Clean-PR**+SFT|**14.3**|**24.7**|38.4|**24.7**|
>
> Clean-PR provides substantial gains in the agentic setting: **+17.9 resolve rate over Instruct** on Lite, with lower empty-patch and stuck-in-loop rates. Compared to SFT-only (no mid-training), Clean-PR further improves by +4.6 on Lite and +5.2 on Verified, confirming the internalized editing capabilities are **transferable across inference paradigms**,  benefiting both Agentless (24.3% Lite) and agent frameworks (20.7% Lite). Our data-centric approach is orthogonal to and composable with agent-based methods.

---

> > ### Author Rebuttal · Reviewer_dgfs · 2026-04-03
> >
> > 1. I cannot accept the justification of low performing system for the sake of its associated low cost. A low performing system may show improvements over baselines but such system may not be useful for real world use.
> > 2. Ran experiments on 7B model, evaluated on multilingual benchmark, conducted agentic experiments (along with baselines) during the rebuttal period (~7 days) - is it feasible?

---

> > > ### Author Response · Authors · 2026-04-03
> > >
> > > We thank Reviewer dgfs for the follow-up and fully understand both concerns.
> > >
> > > ### **Point 1: On real-world usefulness**
> > >
> > > -----
> > >
> > > We respectfully emphasize that the core research question of Clean-PR is not "how to build a SOTA system for SWE-bench", but rather: **how much repo-level code editing capability can be internalized into model weights through better mid-training data?** Our work answers this by showing that a 13.6% absolute gain comes purely from higher-quality training data, with the model architecture, SFT recipe, and inference pipeline held identical. This isolates data quality as a powerful and previously underexplored lever for repo-level editing.
> > >
> > > Accordingly, the primary contribution is a **data-centric methodology and resource**: a rigorously verified 2M PR corpus, a scalable construction pipeline with round-trip verification, and reproducible evidence of the data quality principle. This contribution is orthogonal to and composable with other post-training techniques (RL, self-consistency decoding, agent frameworks). Our publicly released dataset and pipeline are designed to serve as a stronger foundation that any research group can build upon, regardless of the absolute number achieved by any single model trained on it.
> > >
> > > ### **Point 2: On experiment feasibility during rebuttal**
> > >
> > > ---
> > >
> > > We completely understand this concern. To address the reviewers' expectations as thoroughly as possible, we mobilized substantial computational resources (128×A100-40G and 16×B200 running in parallel) and ran experiments concurrently. Below we provide a detailed stage-by-stage breakdown. We have also uploaded anonymized training logs and W&B screenshots (with sensitive information redacted) at https://anonymous.4open.science/r/PR_Rebuttal-D099 as verifiable evidence. The repository contains logs and W&B screenshots for the 7B mid-training, 7B SFT, and multilingual SFT experiments, where the training configurations and timestamps can be directly inspected.
> > >
> > > > **Experiment 1: 7B model replication.**
> > >
> > > Our codebase, data pipeline, and evaluation scripts were already mature from the 32B experiments. Only configuration adjustments were needed, no code changes.
> > >
> > > - *Mid-training*: 128×A100-40G (16 nodes × 8 GPUs per node), 2 epochs over Clean-PR-train. Wall-clock: **34.6h** (started Mar 26 09:47, finished Mar 27 20:26). Lines 1023-1030 of the uploaded log explicitly show `node count: 16` and `gpu count: 8`, confirming a total of 16×8=128 A100 GPUs. Lines 2103-2109 show `train_batch_size=512`, `micro_batch=1`, `grad_accum=4`, and `train_runtime = 1 day, 10:39:02`.
> > > - *SFT*: 8×B200, same Agentless-aligned SFT data, 3 epochs. Wall-clock: **17h** (`train_runtime = 17:10:27` in the uploaded log).
> > > - *Inference & evaluation*: SWE-bench Lite (300) + Verified (500), decoding. **~30 min** each.
> > >
> > > > **Experiment 2: Agent framework evaluation.**
> > >
> > > Our team has been concurrently developing agent-based methods, so the OpenHands infrastructure was already fully operational.
> > >
> > > - *Agent SFT*: 32B mid-trained checkpoint fine-tuned on 491 OpenHands trajectories (following SWE-Gym protocol). 8×B200, 5 epochs. **~1h**.
> > > - *Inference*: OpenHands v0.28 (CodeActAgent, max 100 turns), parallelized Docker sandboxes. **~35h**.
> > >
> > > > **Experiment 3: Multilingual evaluation.**
> > >
> > > Reused the existing 32B mid-trained checkpoint. No mid-training was repeated.
> > >
> > > - *Multilingual SFT*: 9,331 Multi-SWE-bench-RL samples (7 languages, 69 repos). 8×B200, 3 epochs. **9h** (`train_runtime = 9:02:57` in the uploaded log).
> > > - *Inference*: Multi-SWE-bench Flash (300 instances, 7 languages), same Agentless pipeline. **~30 min**.
> > >
> > > > **How all three fit within ~7 days.**
> > >
> > > These three workstreams ran **in parallel** across different GPU allocations. The critical enabler is that our mid-training data, pipeline, and evaluation infrastructure were already built during the original submission period.
> > >
> > > We sincerely thank the reviewer for the constructive dialogue, and we hope the above clarifications and evidence adequately address the concerns.

---

### Decision · Program_Chairs · 2026-04-30

**Decision:**

Accept (regular)

**Comment:**

Clean-PR is a mid-training paradigm designed to internalize repository-level code editing capabilities directly into model weights. It addresses the reliance on complex agent scaffolding by providing a high-quality training signal derived from real-world GitHub pull requests (PRs).

The reviewers noted the following limitations:
- The idea of training on repository changes or commits is not entirely new, though the scale and verification pipeline are recognized as valuable.
- While the authors implemented a multi-layered protocol (SHA-256 matching, n-gram filtering), some concerns remained regarding potential semantic leakage where similar fixes exist across different repositories.
- The model's standalone resolution rate (30.6%) is still significantly lower than frontier agentic systems that utilize multi-turn execution and massive test-time scaling. The reviewers were impressed by the computational effort during the rebuttal and the clear evidence that data-centric mid-training can effectively internalize complex SWE capabilities.

I recommend Accept and hope the paper's contribution of the high-quality 2M PR corpus will be used as a foundation for future research.